# Ubiquitylation of MLKL at lysine 219 positively regulates necroptosis-induced tissue injury and pathogen clearance

Laura Ramos Garcia [1✉], Tencho Tenev[1], Richard Newman[1], Rachel O. Haich [2], Gianmaria Liccardi[1,3], Sidonie Wicky John[1], Alessandro Annibaldi[1,4], Lu Yu [5], Mercedes Pardo [5], Samuel N. Young[6], Cheree Fitzgibbon[6], Winnie Fernando[1], Naomi Guppy[1], Hyojin Kim[1], Lung-Yu Liang[6,7], Isabelle S. Lucet [6,7], Andrew Kueh[6], Ioannis Roxanis[1], Patrycja Gazinska[1], Martin Sims[8], Tomoko Smyth[8✉], George Ward[8], John Bertin[9,10], Allison M. Beal[9], Brad Geddes[9], Jyoti S. Choudhary [5], James M. Murphy [6,7], K. Aurelia Ball [11], Jason W. Upton [2] & Pascal Meier [1✉]

Necroptosis is a lytic, inflammatory form of cell death that not only contributes to pathogen clearance but can also lead to disease pathogenesis. Necroptosis is triggered by RIPK3-mediated phosphorylation of MLKL, which is thought to initiate MLKL oligomerisation, membrane translocation and membrane rupture, although the precise mechanism is incompletely understood. Here, we show that K63-linked ubiquitin chains are attached to MLKL during necroptosis and that ubiquitylation of MLKL at K219 significantly contributes to the cytotoxic potential of phosphorylated MLKL. The K219R MLKL mutation protects animals from necroptosis-induced skin damage and renders cells resistant to pathogen-induced necroptosis. Mechanistically, we show that ubiquitylation of MLKL at K219 is required for higher-order assembly of MLKL at membranes, facilitating its rupture and necroptosis. We demonstrate that K219 ubiquitylation licenses MLKL activity to induce lytic cell death, suggesting that necroptotic clearance of pathogens as well as MLKL-dependent pathologies are influenced by the ubiquitin-signalling system.

[1] The Breast Cancer Now Toby Robins Research Centre, The Institute of Cancer Research, London, UK. [2] Department of Biological Sciences, Auburn University, Auburn, AL, USA. [3] Institute of Biochemistry I, Medical Faculty, Joseph-Stelzmann-Str. 44, University of Cologne, Cologne, Germany. [4] Center for Molecular Medicine Cologne (CMMC), Cologne, Germany. [5] Functional Proteomics Group, The Institute of Cancer Research, London, UK. [6] Walter and Eliza Hall Institute of Medical Research, Parkville, VIC, Australia. [7] Department of Medical Biology, University of Melbourne, Parkville, VIC, Australia. [8] Astex Pharmaceuticals, Cambridge, UK. [9] Innate Immunity Research Unit, GlaxoSmithKline, Collegeville, PA, USA. [10] Immunology and Inflammation Research Therapeutic Area at Sanofi, Cambridge, MA, USA. [11] Department of Chemistry, Skidmore College, Saratoga Springs, NY, USA. ✉email: Laura.ramosgarcia@icr.ac.uk; tomoko.smyth@astx.com; pmeier@icr.ac.uk

Necroptosis is a caspase-independent, lytic form of programmed cell death that is triggered as host-defence mechanism for the elimination of pathogen-infected cells[1]. Infection-induced necroptosis not only deprives pathogens of a replicative niche, but also promotes an adaptive immune response[2]. Evasion of cell death is therefore crucial for successful propagation and dissemination of many pathogens, including intracellular bacteria and viruses[1]. Although necroptosis is an effective mechanism of pathogen clearance, unrestrained lytic cell death can also lead to tissue injury and severe illness, such as acute respiratory distress syndrome, inflammatory diseases and neurodegenerative diseases[3–5].

The necroptotic pathway can be triggered by certain viruses, pathogen components, such as LPS, as well as by death ligands, including TNF and TRAIL[6,7]. Among these, TNF-mediated activation of TNF receptor 1 is the best-understood initiator of necroptosis signalling. Necroptosis depends on the activation of the mixed lineage kinase domain-like (MLKL) pseudokinase by receptor-interacting protein kinase 3 (RIPK3). RIPK3-mediated phosphorylation of mouse MLKL at S345 triggers a conformational change that facilitates the oligomerization, translocation to and disruption of cellular membranes, ultimately causing lytic cell death[8–11]. RIPK3 is activated by RHIM-domain containing proteins, such as RIPK1, TRIF or ZBP1[7]. Importantly, the cysteine protease caspase-8 suppresses necroptotic cell death by cleaving and inactivating proteins that promote necroptosis, including RIPK1, RIPK3 and CYLD[12]. Thus, necroptosis predominantly occurs when caspase-8 activity is low, absent or suppressed, such as in response to viral inhibitors[13,14], pharmacological inhibition[15] and genetic or epigenetic loss[6,16].

Mechanistically, MLKL functions as a molecular switch that exposes its membrane permeabilizing N-terminal 4-helix bundle (4HB) upon RIPK3-mediated phosphorylation in MLKL's activation loop[8,11,17–19]. Phosphorylation in the activation loop results in the destabilisation of a critical hydrogen bond. Breakage of this bond results in the rearrangement of the pseudoactive site of the pseudokinase domain of MLKL[8], which in turn is transmitted to the 4HB domain by two brace helices that link the pseudokinase domain to the 4HB[20]. These helices also provide an interface for oligomerization of MLKL, a requirement for its killing function[20].

While phosphorylation of MLKL constitutes a key signalling step, and is a hallmark of necroptotic signalling, the precise mechanism of MLKL activation is incompletely understood. Here, we demonstrate that endogenous MLKL is readily conjugated with K63-linked Ub-signalling chains during necroptosis, and that ubiquitylation of K219 plays a critical role for the execution of MLKL-dependent cell death. Accordingly, Mlkl[K219R/K219R] knock-in mice are protected from necroptosis-induced tissue injury. Moreover, Mlkl[K219R] mutant cells are protected from necroptosis triggered by TNF or murine cytomegalovirus (MCMV), and fail to restrict viral growth. Mechanistically, we find that K219 ubiquitylation contributes to optimal oligomerisation at cellular membranes, facilitating the rupture of the plasma membrane and lytic cell death. Taken together, our observations are consistent with the notion that K219 ubiquitylation enhances the potential of phosphorylated MLKL to permeabilize the plasma membrane.

## Results

### MLKL is ubiquitylated during necroptosis

The molecular mechanism that regulates the cytotoxic potential of MLKL is not fully understood. To study the regulatory mechanism of MLKL-mediated necroptosis, we monitored the ubiquitylation status of MLKL in cells exposed to necroptotic triggers. While treatment with TNF/SMAC mimetic (SM)/z-VAD-FMK (TSZ) caused time- and

RIPK1-dependent necroptosis in human colorectal cancer HT-29 cells, mouse dermal fibroblasts (MDFs) and murine L929 cells, TSZ also triggered ubiquitylation of endogenous MLKL in all these cell types (Fig. 1a–d and Supplementary Fig. 1a–c). Treatment with the non-specific deubiquitylase USP21 completely removed the smearing pattern of MLKL, demonstrating that MLKL is modified by Ub adducts in response to TSZ (Fig. 1c and Supplementary Fig. 1d). Intriguingly, the extent of MLKL ubiquitylation correlated with the extent of necroptosis. Accordingly, phosphorylation and ubiquitylation of MLKL occurred with a similar kinetics (Fig. 1c, d), and slowing down the kinetics of necroptosis also delayed the appearance of the phosphorylated and ubiquitylated forms of MLKL (Supplementary Fig. 1a–c). We also found that ubiquitylated MLKL was phosphorylated (Fig. 1d, second panel). Ubiquitylation of MLKL not only occurred in response to TNF-induced necroptosis, but also upon necroptosis triggered by TRAIL (Fig. 1e, f). This suggests that ubiquitylation of MLKL occurs in response to various necroptotic signalling events. Importantly, MLKL was not ubiquitylated during TNF-induced apoptosis (Fig. 1g, h). Taken together, these results indicate that MLKL is ubiquitylated in response to necroptotic stimuli.

### Ubiquitylation of MLKL requires RIPK3-mediated phosphorylation

RIPK3 activates mouse MLKL via phosphorylation at S345, which causes a conformational change of MLKL that enables its trimerization and subsequent translocation to the plasma membrane where MLKL trimers form higher-order polymers, which permeabilize the plasma membrane (Fig. 2a)[9,11,20]. Next, we assessed whether RIPK1 > RIPK3 signalling is required for MLKL ubiquitylation. We found that pharmacological inhibition of RIPK1 and RIPK3, or genetic deletion of RIPK3 not only blocked necroptosis (Supplementary Fig. 2a, b), but also abrogated ubiquitylation of MLKL (Fig. 2b, c). These data demonstrate that MLKL ubiquitylation is highly dependent on the kinase activity of RIPK3. To corroborate this notion, we tested whether RIPK3-mediated phosphorylation of S345 is required for MLKL ubiquitylation. To this end, we reconstituted Mlkl[−/−] MDFs with wild type (WT) or a mutant form of MLKL (MLKL[S345A]) that cannot be phosphorylated by RIPK3, and hence fails to trigger necroptosis (Supplementary Fig. 2c)[18]. While WT MLKL was readily ubiquitylated upon treatment with TSZ, ubiquitylation of MLKL[S345A] was significantly suppressed (Fig. 2d). We observed that the appearance of the prominent, lower molecular weight ubiquitylation smearing pattern of MLKL was dependent on MLKL phosphorylation at S345. In contrast, the faint, higher molecular weight ubiquitylation smearing pattern occurred irrespective of MLKL phosphorylation at S345 (Fig. 2d and Supplementary Fig. 2d). This result is consistent with the notion that the active, phosphorylated form of MLKL is ubiquitylated.

We, therefore, assessed whether the ubiquitylated form of MLKL is also incorporated into oligomers, and, therefore, might influence plasma membrane disruption. Since intermolecular disulfide bonds stabilise trimeric, active MLKL[9,10,21], we used non-reducing conditions to evaluate MLKL oligomerisation. While the treatment with TSZ caused trimerization of MLKL (Supplementary Fig. 2e)[9,21,22], we found that MLKL oligomers were also ubiquitylated (Fig. 2e). Accordingly, incubation with USP21 abrogated the Ub smearing pattern, leading to the appearance of oligomeric, as well as monomeric forms of MLKL (Fig. 2f). Finally, adding a FLAG tag on MLKL's N-terminus, which prevents higher-order assembly at the plasma membrane[11], did not affect MLKL ubiquitylation (Supplementary Fig. 2f). Together, these data indicate that both phosphorylated monomers and oligomers are ubiquitylated, suggesting that ubiquitylation of P-MLKL is an early

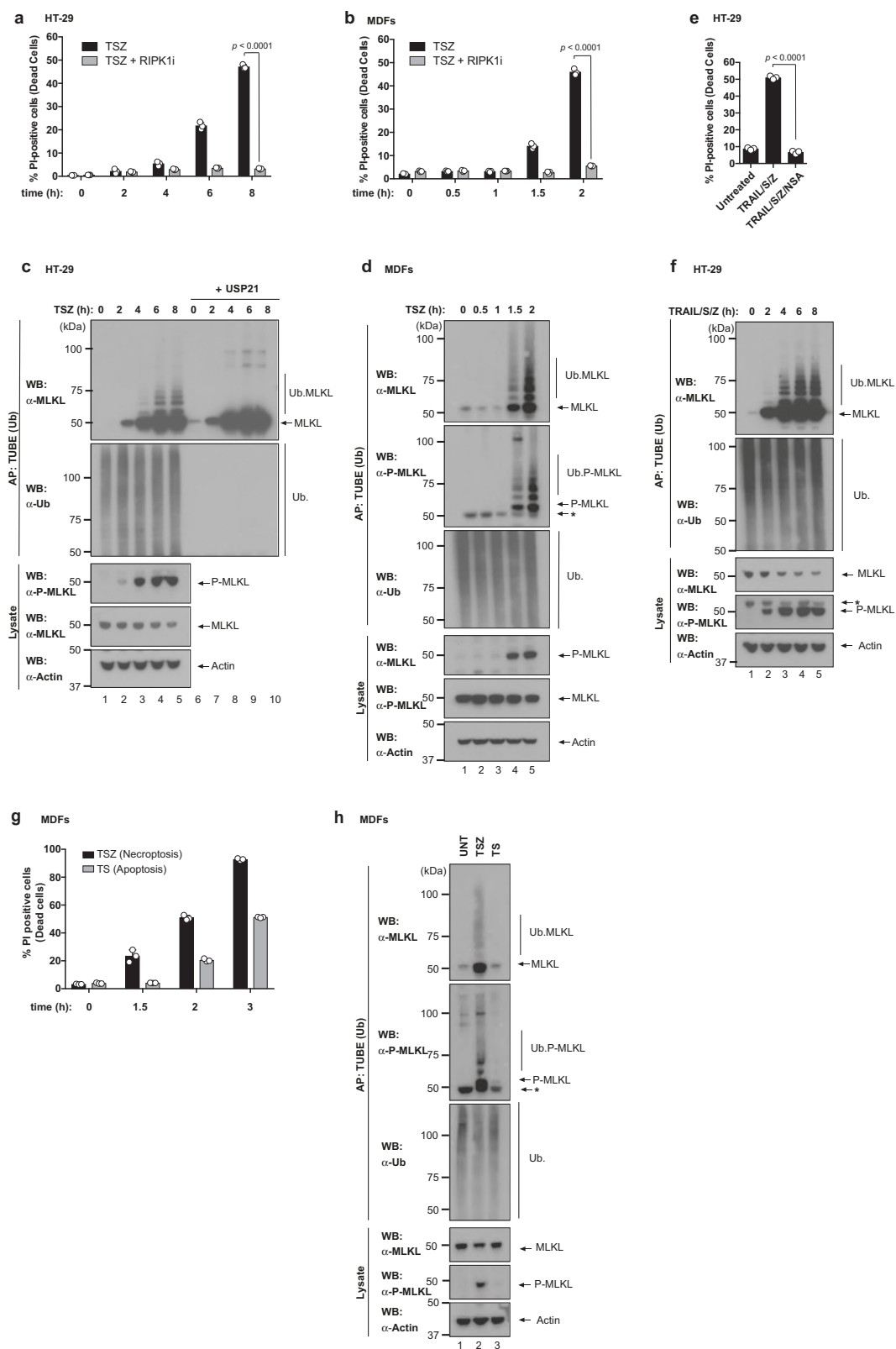

event, and occurs prior to its oligomerisation and membrane translocation.

**MLKL is ubiquitylated prior to plasma membrane localisation**. Following oligomerisation, active MLKL translocates to the plasma membrane where it accumulates to induce cell death[9,10,23–25]. To address whether ubiquitylation of MLKL occurs prior to plasma membrane trafficking, we used in situ proximity ligation assay (PLA) and subcellular fractionation approaches. Using primary antibodies against MLKL and Ub, we observed that treatment with TSZ caused prominent MLKL:Ub PLA speckles, while no such speckles were detected under untreated control conditions (Fig. 3a). Interestingly, these PLA speckles localised to the cytosol, suggesting that MLKL is ubiquitylated in the cytosol prior to its translocation

**Fig. 1 Endogenous MLKL is ubiquitylated during necroptosis. a** Quantification of propidium iodide positive (PI+) HT-29 cells upon treatment with TNF (10 ng/ml), SM-164 (100 nM) and z-VAD-FMK (20 µM; TSZ) in the presence or absence of RIPK1 inhibitor (RIPK1i GSK'963, 100 nM) for the indicated times. **b** Quantification of PI+ mouse dermal fibroblasts (MDF) treated as in **a**. **c** Tandem ubiquitin-binding entities (TUBE) affinity purification (AP) of ubiquitylated proteins from HT-29 cells treated with the indicated agents for the indicated timepoints. Prior to elution from the beads, samples were split in two and incubated with or without 2 µM USP21. The presence of MLKL ubiquitylation was determined by immunoblot analysis of the eluate using α-MLKL antibody. **d** TUBE AP of ubiquitylated proteins from MDFs treated with the indicated agents for the indicated timepoints. * refers to non-specific bands. **e** Quantification of PI+ HT-29 cells upon treatment with TRAIL (50 ng/ml), SM-164 (100 nM) and z-VAD-FMK (20 µM; TRAIL/S/Z) for 8 h in the presence or absence of MLKL inhibitor necrosulfanamide (NSA). **f** TUBE AP of ubiquitylated proteins from HT-29 cells treated with TRAIL/S/Z. * refers to non-specific bands. **g** Quantification of PI+ MDF cells treated with TNF and SM-164 in presence or absence of z-VAD-FMK to induce necroptosis (TSZ) or apoptosis (TS), respectively for the indicated times. **h** TUBE AP of ubiquitylated proteins from MDFs treated to induce necroptosis (TSZ, 2 h) or apoptosis (TS, 3 h). * refers to non-specific bands. In **a**, **b**, **e** and **g**, $n = 3$ wells/group. The results are representative of those from two independent experiments with three technical replicates each. Data are presented as mean with error bars indicating standard deviation (SD). Statistical analysis shown was calculated by two-way ANOVA with Sidak's multiple comparison test in **a**, **b** and one-way ANOVA with Sidak's multiple comparison test in **e**. Source data are provided as a Source data file. **c**–**f**, **h** are representative of ≥1 biological replicates.

to the plasma membrane. This was confirmed using subcellular fractionation approaches to separate the cytosolic content (C) from the membrane-containing fraction (M). By performing cellular fractionation followed by Ub enrichment, we found that TSZ treatment caused the appearance of ubiquitylated MLKL in the cytosolic and membrane-enriched fractions at 2 h (Fig. 3b). As time progressed, more ubiquitylated MLKL accumulated in the membrane-enriched fraction (Fig. 3b, compare lanes 9 with 15). Together, these results support the notion that MLKL is predominantly ubiquitylated in the cytosolic compartment before its translocation to the plasma membrane.

Next, we used a variety of Ub-linkage selective affinity reagents to determine the Ub chain type that is conjugated to MLKL. Using these reagents, we found that necroptotic stimuli triggered prominent K63-linked ubiquitylation of MLKL in MDFs and HT-29 (Fig. 3c, d). Under these conditions, no M1- or K48-linked chains were detectable. The presence of K63-linked Ub chains suggest that ubiquitylation of MLKL serves a signalling function and does not trigger its proteasomal degradation.

**Endogenous MLKL is ubiquitylated on K51, K77, K172 and K219.** To study the role of MLKL ubiquitylation in necroptosis signalling, we next identified the Ub acceptor lysine residues of MLKL. To this end, we used a quantitative mass spectrometry-based approach coupled to immunoprecipitation of K-ε-GG Ub-remnant peptides (Fig. 4a). This identified a TSZ-dependent increase in the abundance of site-specific ubiquitylation events for several proteins of the TNFR1 signalling pathway, including RIPK1, RIPK3, Casp8 and MLKL (Fig. 4a). Examination of the K-ε-GG Ub-remnant peptides of endogenous MLKL revealed TSZ-dependent ubiquitylation of K51, K77, K172 and K219 (Fig. 4b, c), in all three biological replicates. The ubiquitylated residues mapped to the 4HBD (K51 and K77), the second brace helix (K172) and the pseudokinase domain (K219). K219 is of particular interest as this residue forms a hydrogen bond with Q343 of the activation loop under resting condition (Fig. 4c)[8]. While the hydrogen bond between K219 and Q343 locks MLKL into an inactive conformation under resting conditions, RIPK3-medidated phosphorylation at S345 triggers a conformational change that destabilises this bond and re-arranges the pseudoactive site, leading to 'activation' of MLKL[8]. Thus, following phosphorylation at S345, the ε-amino group of K219 becomes available for ubiquitylation. This explains why only phosphorylated, active MLKL is subject to ubiquitylation. This also raises the possibility that ubiquitylation at K219 might stabilise the activated form of MLKL, and hence contributes to the necroptotic potential of MLKL.

**Ubiquitylation at K219 contributes to the cytotoxic potential of MLKL.** To examine the importance of MLKL ubiquitylation, we reconstituted $Mlkl^{-/-}$ MDFs with various Ub acceptor lysine mutants. To generate MLKL that cannot be ubiquitylated at K219, but resides in an inactive conformation, we exchanged K219 to arginine (R). Since Arg has a similar chemical structure to Lys, K219R is predicted to maintain the hydrogen bond with Q343 under resting conditions (Fig. 5a). Accordingly, MLKL$^{K219R}$ behaved comparably to MLKL$^{WT}$ under resting conditions, and their expression was well tolerated and non-cytotoxic in the absence of necroptotic stimuli (Fig. 5b). This was in contrast to MLKL$^{K219M}$, which carries a K219 to M substitution mutation that disrupts the hydrogen bond with Q343[8], and which was cytotoxic (Fig. 5a, b)[8]. Intriguingly, following treatment with TSZ, MLKL$^{K219R}$ was significantly less potent in inducing necroptosis than WT MLKL (Fig. 5c). The difference in necroptotic potency was not caused by differences in their protein stability or expression levels, since MLKL$^{K219R}$ and MLKL$^{WT}$ were expressed to similar levels and exhibited comparable protein thermal stability profiles (Supplementary Fig. 3a, b). This strongly suggests that ubiquitylation of K219 contributes to the cytotoxic potential of MLKL. Of note, K219 of mouse MLKL is conserved in human MLKL (hMLKL), and corresponds to K230. Intriguingly, hMLKL K230 was found to be ubiquitylated in a whole ubiquitinome-wide analysis[26], and is mutated in cancer[19]. Since mutation of K230 renders hMLKL less potent[19], this suggests that, like in the murine setting, ubiquitylation of K230 might contribute to hMLKL's killing function.

In contrast to K219, mutating the Ub acceptor lysines K51 and K77, which are located in the 4HBD of MLKL (MLKL$^{K51R,K77R}$), had no effect in MLKL's killing potential, neither alone nor in combination (Fig. 5c and Supplementary Fig. 3b). Further, the triple mutant MLKL$^{K51R,K77R,K219R}$ had the same reduced killing potential as the single mutant MLKL$^{K219R}$. Mutating the acceptor K172 to R (MLKL$^{K172R}$) showed a modest protection (Fig. 5c and Supplementary Fig. 3b). Although this indicates that ubiquitylation of the brace (K172) may contribute to the killing potential of MLKL, our data suggest that K219 is the predominant site influencing necroptosis. Intriguingly, previous work indicates that MLKL$^{K219M}$ is less cytotoxic than MLKL$^{Q343A}$, even though both mutations disrupt the hydrogen bond between K219 and Q343[8]. Since MLKL$^{Q343A}$ retains the K219 ubiquitylation site, it is possible that MLKL$^{Q343A}$ is more toxic than MLKL$^{K219M}$ because it profits from pro-necroptotic ubiquitylation of K219.

Next, we tested the contribution of K219 ubiquitylation on the cytotoxic potential of a constitutively active mutant form of MLKL (MLKL$^{S345D}$). Induced expression of MLKL$^{S345D, K219R}$ was significantly less potent in inducing necroptosis than

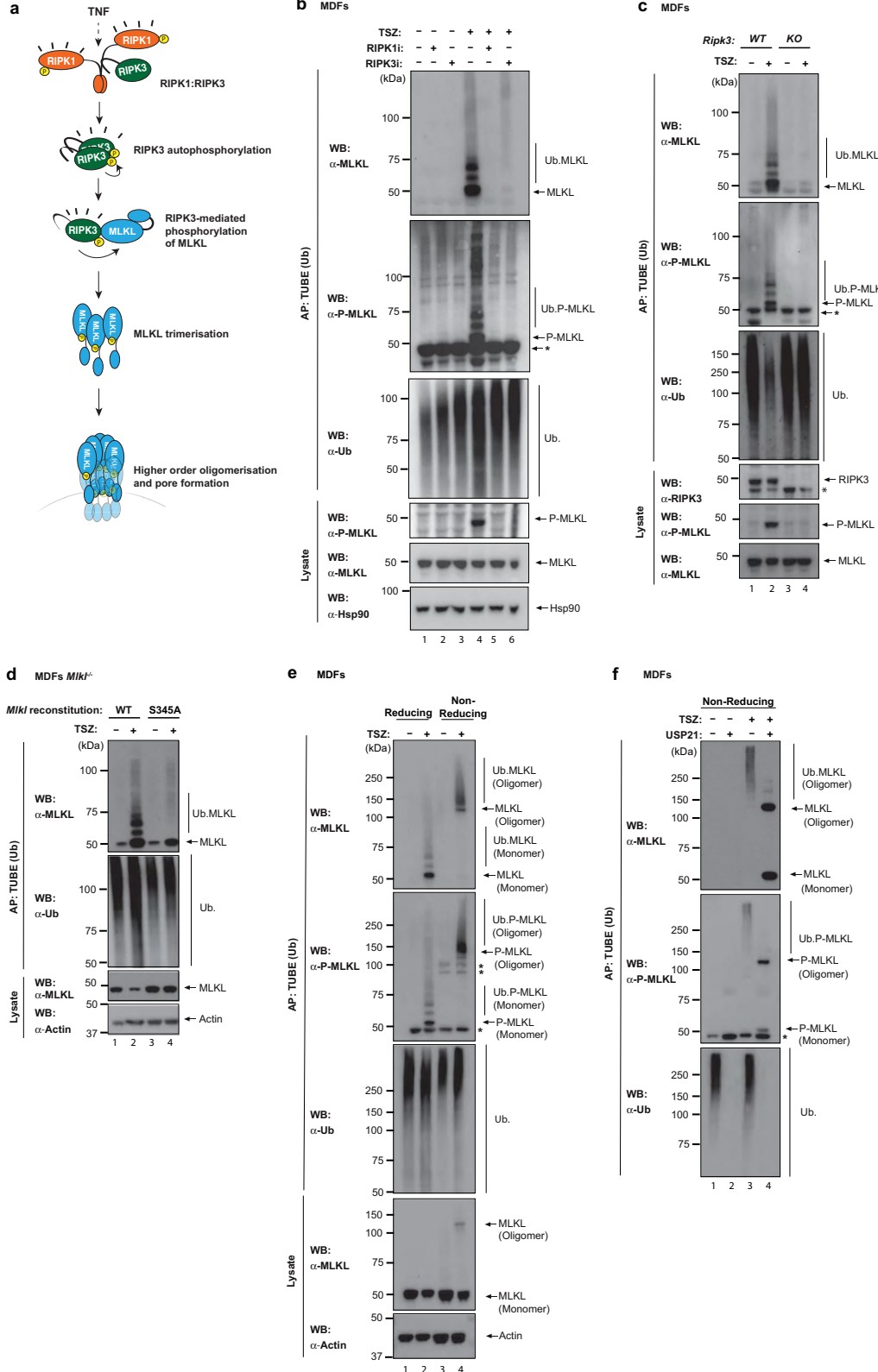

MLKL[S345D] (Fig. 5d and Supplementary Fig. 3c). This further demonstrates that ubiquitylation of K219 contributes to the pronecroptotic potential of MLKL.

To help explain the mechanism by which ubiquitylation of K219 enhances cell death, we performed extensive all-atom molecular dynamics (MD) simulations of unmodified MLKL (WT), MLKL phosphorylated at S345 (S345phos), MLKL both phosphorylated at S345 and monoubiquitylated at K219 (K219ub S345phos), starting from the crystal structure of mMLKL (PBD: 4BTF)[8]. Although the hydrogen bond between K219 and Q343 was formed in the crystal structure, the conformation of the 4HB domain resembled a more open, active conformation. We found that even in the WT simulations, the hydrogen bond was not maintained throughout the simulations (present only in 14% of

**Fig. 2 Ubiquitylation of MLKL requires RIPK3-mediated phosphorylation. a** Schematic depicting the key steps in necroptosis signalling. TNF can drive RIPK1-mediated activation of RIPK3. Active RIPK3 subsequently phosphorylates MLKL to induce MLKL activation. Phosphorylated MLKL trimerises and translocates to membranes where it forms higher-order polymers inducing lytic cell death. **b** Tandem ubiquitin-binding entities (TUBE) affinity purification (AP) of ubiquitylated proteins from mouse dermal fibroblasts (MDF) treated with TNF (10 ng/ml), SM-164 (100 nM) and z-VAD-FMK (20 μM; TSZ) for 2 h in presence or absence of RIPK1 inhibitor (RIPK1i GSK′963, 100 nM) or RIPK3 inhibitor (RIPK3i GSK′843 2 μM). **c** TUBE AP of ubiquitylated proteins from WT and $Ripk3^{-/-}$ MDFs treated for 2 h with TSZ. **d** TUBE AP of ubiquitylated proteins from $Mlkl^{-/-}$ MDFs reconstituted with $Mlkl^{WT}$ or $Mlkl^{S345A}$, and stimulated with TSZ for 2 h. **e** TUBE AP of ubiquitylated proteins from MDFs treated with TSZ. Prior to elution from the beads, the samples were split in two and either eluted with a sample buffer containing β-mercaptoethanol (reducing condition) or without it (non-reducing condition) to enable visualisation of MLKL trimers. **f** TUBE AP of ubiquitylated proteins from MDFs treated with TSZ. Before elution, samples were split in two and treated with USP21 or left untreated. Lysates were eluted in non-reducing conditions to visualise MLKL oligomerization. All data are representative of ≥1 biological replicates. In **b**, **c**, **e** and **f**, * refers to non-specific bands. Source data are provided as a Source data file.

the simulated ensemble), and the 4HB domain conformation sampled both open and closed conformations (Fig. 5e, f and Supplementary Fig. 3d). In the WT simulations, S345 on the activation loop helix is pointed toward the adjacent helix and residue T235, but when S345 is phosphorylated, the phosphate group moves away to interact with solvent instead, disrupting both the structure and orientation of the activation loop helix (Fig. 5e–g and Supplementary Fig. 3d). This conformational change in the pseudokinase domain appears to have an allosteric effect on the 4HB domain conformation, which is more consistently in the open, active conformation in the S345phos simulations (Fig. 5e–h and Supplementary Fig. 3d). We also ran simulations with both S345 phosphorylated and a single Ub molecule covalently attached to the K219 sidechain (Fig. 5e–g). Although K219 is not a surface residue, attachment of the Ub moiety did not significantly disrupt the MLKL fold in our simulations. As in the S345phos simulations, the phosphorylated serine was more solvent exposed and disrupted the structure of the activation loop helix in the K219ub S345phos simulations (Fig. 5e). However, we did not see a significant difference between the activation loop helix position in the K219ub S345phos simulations compared to either the WT or S345phos simulations ($p$ value > 0.4). Surprisingly, the K219ub S345phos simulations sampled on average a 4HB domain more closely packed against the pseudokinase domain than the S345phos simulations, but the position of this domain varied widely within the simulated ensemble (Fig. 5h–i and Supplementary Fig. 3e). Overall MD simulations indicate a conformational change in the activation loop helix and 4HB domain upon S345 phosphorylation, leading to MLKL activation. In addition, the conjugation of Ub to K219 seems to influence the 4HB domain flexibility of P-MLKL, which might be an important feature for MLKL polymerisation.

**$Mlkl^{K219R/K219R}$ cells are protected from TNF- and MCMV-driven necroptosis.** To examine the role of MLKL ubiquitylation at K219 in the regulation of necroptosis, viral infection and tissue injury, we generated knock-in mice expressing $Mlkl^{K219R}$ from the endogenous $Mlkl$ genomic locus (Supplementary Fig. 4a). $Mlkl^{K219R/K219R}$ mice were born at the expected Mendelian frequency, reached adulthood without showing signs of pathology, and were indistinguishable from their $Mlkl^{WT/WT}$ littermates (Supplementary Fig. 4b). To study the effect of the knock-in mutation in necroptosis, we measured TSZ-induced cell death of bone marrow-derived macrophages (BMDMs) and MDFs, isolated from two different pairs of $Mlkl^{WT/WT}$ and $Mlkl^{K219R/K219R}$ mice. $Mlkl^{K219R/K219R}$ BMDMs and MDFs were protected from TSZ-induced necroptosis compared to WT cells (Fig. 6a, b). Under these conditions, $Mlkl^{K219R/K219R}$ cells were as resistant as WT cells treated with RIPK1i (GSK′963) or RIPK3i (GSK′843), suggesting that K219 ubiquitylation is important for the induction of necroptosis (Fig. 6a, b). These data somewhat contrast our findings from the reconstitution system, where we observed only

a partial rescue upon transgene-mediated re-expression of $MLKL^{K219R}$ in $Mlkl^{-/-}$ MDFs. This might be due to the fact that transgene-mediated re-expression of MLKL may not accurately reflect the expression level of endogenous MLKL as observed in cells derived from $Mlkl^{K219R/K219R}$ animals. Of note, the expression levels of RIPK1, RIPK3 and MLKL were indistinguishable in WT and $Mlkl^{K219R/K219R}$ cells (Supplementary Fig. 4c). Further, phosphorylation and activation of MLKL was readily-detected in the lysates of $Mlkl^{K219R/K219R}$ MDFs, even at a time point when no cell death was apparent (2.5 h; Fig. 6c), reinforcing the viewpoint that RIPK3-mediated phosphorylation of MLKL is unaffected in these cells. Analysis of K63-linked ubiquitylation on $Mlkl^{K219R/K219R}$ knock-in cells and $Mlkl^{-/-}$ MDFs reconstituted with $Mlkl^{K51R,K77R,K172R,K219R}$ mutant, showed a reduction in the Ub smear, when compared to the one in $Mlkl^{WT/WT}$ cells (Supplementary Fig. 4d, e), although the persistence of a Ub smear suggests that additional ubiquitylation sites might exist. Taken together, our results show that abolishing MLKL ubiquitylation at K219 protects from MLKL-mediated cell death.

Next, we evaluated whether ubiquitylation at K219 contributes to the formation of higher-order MLKL polymers at the plasma membrane during necroptosis. To this end, we performed Blue Native PAGE (BN-PAGE) on cytosolic and membrane fractions of WT and $Mlkl^{K219R/K219R}$ MDFs. While treatment with TSZ stimulated the formation of large MLKL polymers on membranes of WT cells (Fig. 6d)[11,19], such polymers barely formed in $Mlkl^{K219R/K219R}$ MDFs (Fig. 6d), even though $MLKL^{K219R/K219R}$ was phosphorylated (Fig. 6c and Supplementary Fig. 4d). Our interpretation of these findings is that phosphorylation of MLKL at S345 is not sufficient to trigger necroptosis, and that ubiquitylation of K219 is additionally required for cell death. Our data are consistent with the notion that ubiquitylation at K219 contributes to the cytotoxic potential of MLKL by facilitating the formation of higher-order oligomers at the plasma membrane.

Cell death plays an important role in the defence against certain types of pathogens, such as viruses[1,27]. MCMV triggers both apoptosis and necroptosis in infected cells. However, the MCMV-encoded M36 and M45 proteins inhibit caspase-8 and RIPK3 signalling, respectively, thereby suppressing these antiviral responses[13,14]. In contrast to WT MCMV particles, MCMV variants that carry a mutation in the RHIM domain of M45 (MCMV$^{M45mutRHIM}$), fail to block necroptosis, and cause cell death of the infected cell[13,14]. To explore the role of K219 ubiquitylation in MCMV-induced cell death, we infected cells with MCMV$^{WT}$ and MCMV$^{M45mutRHIM}$. Infection with MCMV$^{M45mutRHIM}$ readily induced cell death in WT MDFs (Fig. 6e). In contrast, $Mlkl^{K219R/K219R}$ MDFs were protected from MCMV-induced necroptosis (Fig. 6e). To further investigate the effect of K219 ubiquitylation on virus growth, we determined the viral titres of infected cells. Consistent with the notion that $Mlkl^{K219R/K219R}$ MDFs failed to clear the MCMV$^{M45mutRHIM}$

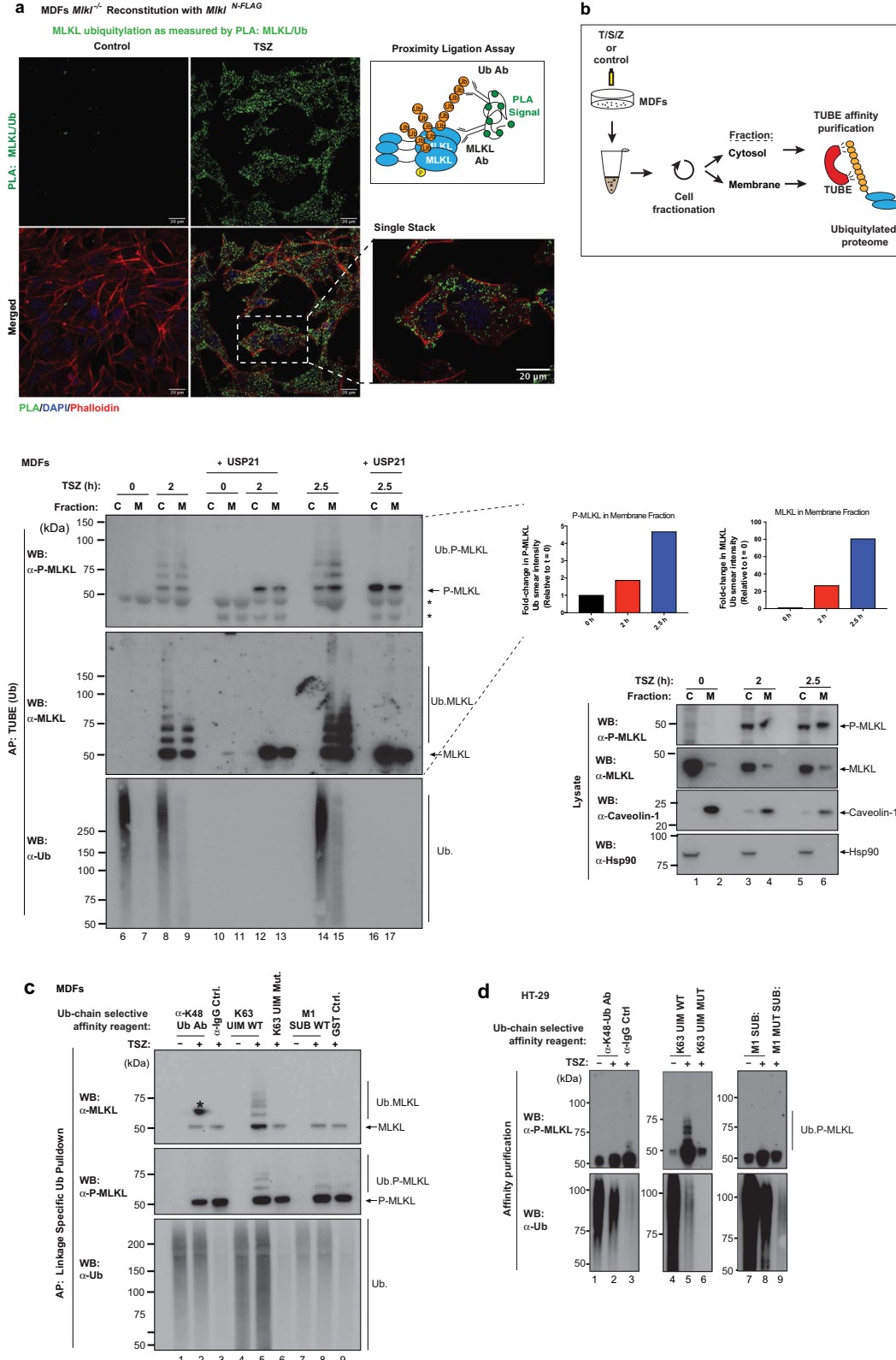

infection via necroptosis, $Mlkl^{K219R/K219R}$ MDFs had a significantly higher viral titre than WT cells 72 h after infection (Fig. 6f). The failure of $Mlkl^{K219R/K219R}$ cells to clear MCMV$^{M45mutRHIM}$ infection phenocopies $Ripk3^{-/-}$, $Zbp1^{-/-}$ and $Zbp1^{ZBDmut}$ infected cells[13,14,28]. Of note, MCMV$^{WT}$ did not induce cell death in either WT or $Mlkl^{K219R/K219R}$ MDFs (Fig. 6e),

and replicated with the same efficiency in both WT and $Mlkl^{K219R/K219R}$ MDFs (Fig. 6f), corroborating that the K219 mutation does not affect viral infection per se. Taken together, these observations are consistent with the notion that ubiquitylation of K219 is required for TNF- and MCMV-induced necroptosis, and attenuation of viral growth.

**Fig. 3 MLKL is ubiquitylated prior to plasma membrane localisation. a** Proximity ligation assay (PLA) of $Mlkl^{-/-}$ mouse dermal fibroblasts (MDF) reconstituted with $Mlkl^{N-FLAG}$. Cells were treated with TNF (10 ng/ml), SM-164 (100 nM) and z-VAD-FMK (20 μM; TSZ) for 2 h. Right panel, schematic depicting the principle of PLA. Antibodies that detect MLKL and ubiquitin (Ub) were utilised. **b** MDFs were stimulated with TSZ for the indicated timepoints and subjected to cellular fractionation as described in the 'Methods' section. The corresponding cytoplasmic (C) and membrane (M) fractions were subjected to a tandem ubiquitin-binding entities (TUBE) affinity purification (AP). The experimental scheme is depicted. Prior to elution from the beads, the samples were split in two and subsequently left untreated or incubated with USP21. * refers to non-specific bands. Quantification of P-MLKL and total MLKL in the membrane fraction is shown. **c** Linkage-specific ubiquitin (Ub) AP in MDF stimulated with TSZ for 2 h. The indicated affinity reagents were used to purify the respective Ub chain types. * refers to a developing artefact. **d** Linkage-specific ubiquitin AP in HT-29 cells treated with TSZ for 6 h. The indicated affinity reagents were used to purify the respective Ub chain types. All results (**a**–**d**) are representative of those from two independent experiments. Source data are provided as a Source data file.

**$Mlkl^{K219R/K219R}$ knock-in mice are protected from necroptosis-induced tissue injury.** Next, we tested $Mlkl^{K219R/K219R}$ knock-in mice for their sensitivity to necroptosis-induced tissue injury. Previous work established that subcutaneous injection of SM causes RIPK1-dependent cell death and tissue injury that mimics inflammatory disease seen in genetic models of IAP loss[29]. We used this system in combination with the clinical pan-caspase inhibitor emricasan to evaluate the effect of necroptosis in the skin of WT and $Mlkl^{K219R/K219R}$ knock-in animals. $Mlkl^{-/-}$ animals served as controls. Subcutaneous injection of the clinical IAP antagonist ASTX660 in combination with emricasan (AE treatment) caused substantial ulceration of the skin of $Mlkl^{WT/WT}$ animals at 72 h (Fig. 7a–c). In contrast, $Mlkl^{K219R/K219R}$ knock-in animals were significantly protected from AE-induced tissue injury, which was comparable to $Mlkl^{-/-}$ animals (Fig. 7a–i). The WT group treated with AE achieved the highest histological multivariant lesion score (HLS; Supplementary Fig. 5a) of mean 285 (range 242–356), while $Mlkl^{-/-}$ or knock-in animals reached an HLS mean 55.5 (range 15.20–160) and mean 82.50 (range 36.10–146.70), respectively (Fig. 7c). Subcutaneous injection of AE in WT animals caused ulceration of the epidermis with dense collagenous fibrous tissue formation in underlying dermis, that appeared extending to the hypodermis (Fig. 7b–i and Supplementary 6a). WT animals also showed clear evidence of cellular necrosis, including the presence of nuclear dust mostly at superficial or mid-dermis (Supplementary Fig. 6a magnification 1), together with scattered necrotic cells within the deep region of the dermis and hypodermis (Supplementary Fig 6b, magnification 2). Necrotic cells were also observed in the root-sheet epithelium of hair follicles located at the deeper hypodermis of AE-injected WT animals (Supplementary Fig. 6b, magnification 3). In contrast to WT animals, subcutaneous injection of AE into $Mlkl^{K219R/K219R}$ and $Mlkl^{-/-}$ animals did not result in any of the aforementioned features (Fig. 7b). Accordingly, $Mlkl^{K219R/K219R}$ and $Mlkl^{-/-}$ animals exhibited an epidermal layer without ulceration (Fig. 7b). Moreover, there was no evidence of necrosis in $Mlkl^{K219R/K219R}$ and $Mlkl^{-/-}$ mice (Fig. 7b). Although $Mlkl^{K219R/K219R}$ were significantly protected from the effects of subcutaneous injection of AE, such animals did show some irregular epidermis (Fig. 7d) due to thickening of the epidermis (Fig. 7e). Nevertheless, when compared to WT animals, subcutaneous injection of AE only caused relatively minor effects in $Mlkl^{K219R/K219R}$ animals (Fig. 7i). Importantly, quantification of immune cells within the spleens of $Mlkl^{K219R/K219R}$ knock-in mice and $Mlkl^{-/-}$ mice excluded any basal alteration within their immune landscape (Supplementary Fig. 5b). Lastly, to test whether subcutaneous injection of AE-induced RIPK1-dependent necroptosis, mice were fed chow-containing RIPK1 inhibitor (Ripk1i GSK′547) for 7 days prior to injection. As shown in Supplementary 7a–i, RIPK1 inhibition conferred significant protection in comparison to animals fed on a control diet. Together, these data demonstrate that subcutaneous injection of AE induces a RIPK1-driven necroptotic cell death phenotype, which is lost in $Mlkl^{K219R/K219R}$ mice. This suggests that conjugation of Ub to K219 contributes to necroptosis-mediated tissue damage of the skin.

## Discussion

Necroptosis is a lytic, inflammatory form of cell death that not only contributes to pathogen clearance[30], but can also lead to tissue injury and disease pathogenesis if aberrantly activated[31–34]. Despite the identification of key players of the necroptosis machinery, surprisingly little is known about how MLKL is activated and how this potentially catastrophic event is regulated. Here, we show that endogenous MLKL is ubiquitylated during necroptosis, and that ubiquitylation at K219 plays a critical role for the execution of MLKL-dependent cell death. Moreover, we demonstrate that K219 ubiquitylation is required for MLKL-dependent pathogenesis in vivo, and contributes to MLKL-mediated viral clearance. Mechanistically, we provide evidence that ubiquitylation of MLKL facilitates the formation of higher-order polymers required for membrane rupture and lytic cell death. Therefore, K219 ubiquitylation regulates the cytotoxic potential of MLKL.

We find that MLKL is ubiquitylated with K63-linked Ub chains during necroptosis, but not apoptosis. Accordingly, ubiquitylated forms of MLKL were only detected following treatment with necroptotic stimuli, such as TSZ and TRAIL/S/Z, but not under resting conditions or during TS-induced apoptosis. The prominent low molecular weight ubiquitylation events on MLKL were dependent on RIPK3-mediated phosphorylation of MLKL at S345. Interfering with MLKL phosphorylation, either via pharmacological inhibition of RIPK1 and RIPK3, or expression of a phospho-mutant form of MLKL (MLKL$^{S345A}$), abrogated MLKL phosphorylation and Ub modifications of low molecular weight. Intriguingly, while the low molecular weight ubiquitylation events were phospho-dependent, higher molecular weight modifications appeared to be slightly elevated upon inhibition of S345 phosphorylation. Consistent with the requirement of MLKL phosphorylation for its ubiquitylation, we found that MLKL ubiquitylation directly correlated with the extent of MLKL phosphorylation and necroptosis.

The mechanism of MLKL activation helps to explain why only phosphorylated MLKL is ubiquitylated at K219. In its non-phosphorylated state, MLKL adopts a closed, inactive configuration. In mice, this inactive configuration is maintained by a critical hydrogen bond between K219 and Q343, which is located in the helix that forms the activation loop[11]. Upon phosphorylation of MLKL at S345, the K219::Q343 hydrogen bond is destabilised, which triggers a conformational switch that converts MLKL from a dormant cytoplasmic protein into oligomers that translocate to, and permeabilise, the plasma membrane to kill cells. Phospho-dependent breakage of the K219::Q343 hydrogen bond not only results in a conformational change of MLKL, but also frees the ε-amino group of K219 so that it can become available to serve as Ub acceptor. This explains why only the phosphorylated form of MLKL is ubiquitylated. It also explains why MLKL is exclusively

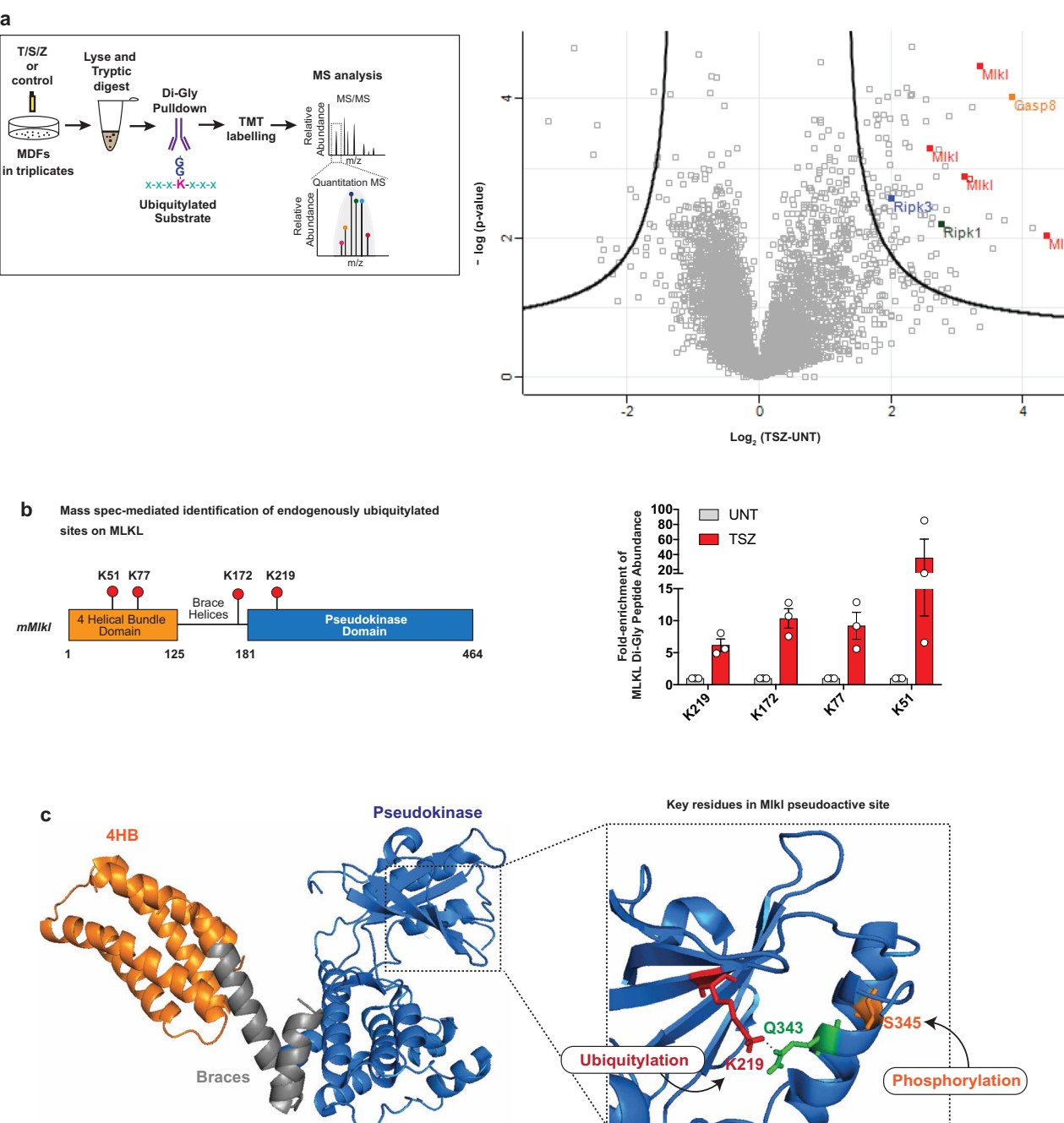

**Fig. 4 Endogenous MLKL is ubiquitylated on K51, K77, K172 and K219. a** Mass spectrometry (MS)-based identification of ubiquitylated peptides. Left panel: schematic representation of the experimental design. Mouse dermal fibroblasts (MDF) were treated with TNF (10 ng/ml), SM-164 (100 nM) and z-VAD-FMK (20 μM; TSZ) for 2 h or left untreated. Extracted proteins were digested with trypsin and Ub remnants were enriched using an anti-K-ε-GG antibody. Six samples, corresponding to three biological replicates, were labelled with one of the tandem mass tags (TMT) 10-plex. Subsequently, samples were mixed, subjected to identification by LC–MS/MS and quantified using peak area under curve. Right panel: volcano plot of global peptide abundance showing −log *p* values versus log2 ratio changes between TSZ and untreated control. The three biological replicates were taken into consideration. Peptides corresponding to MLKL, RIPK1, RIPK3 and caspase-8 are indicated. **b** Schematic diagram depicting the domain architecture of murine MLKL and the identified Ub acceptor lysine (K), left panel. The graph depicts fold enrichment of the respective di-Gly peptide abundance. The result is representative of one independent experiment with three replicates for each treatment group. Data are presented as mean ± the standard error of the mean (SEM). **c** Crystal structure of murine MLKL (PBD:4BTF[8]) comprising the 4-helix bundle (4HB) domain (orange), the pseudokinase domain (blue) and the brace helices (grey). Magnification of the pseudoactive site of MLKL showing the hydrogen bond interaction between K219 and Q343. Residue S345 that is phosphorylated by RIPK3 is shown in orange.

ubiquitylated under necroptotic conditions, and not during apoptosis or resting conditions. Similar to RIPK1 (ref. [35]) and RIPK3 (ref. [36]), MLKL is also a client protein of heat shock protein 90 (Hsp90). This suggests that Hsp90 may also be involved in phosphorylation-mediated conformational changes of

MLKL's pseudokinase domain[37–39]. As such, Hsp90 might influence the availability of K219 as an Ub acceptor. Of note, K219 of mouse MLKL is conserved in hMLKL, corresponding to K230. Intriguingly, K230 of hMLKL is mutated in cancer[19], suggesting that this residue is an important regulatory site.

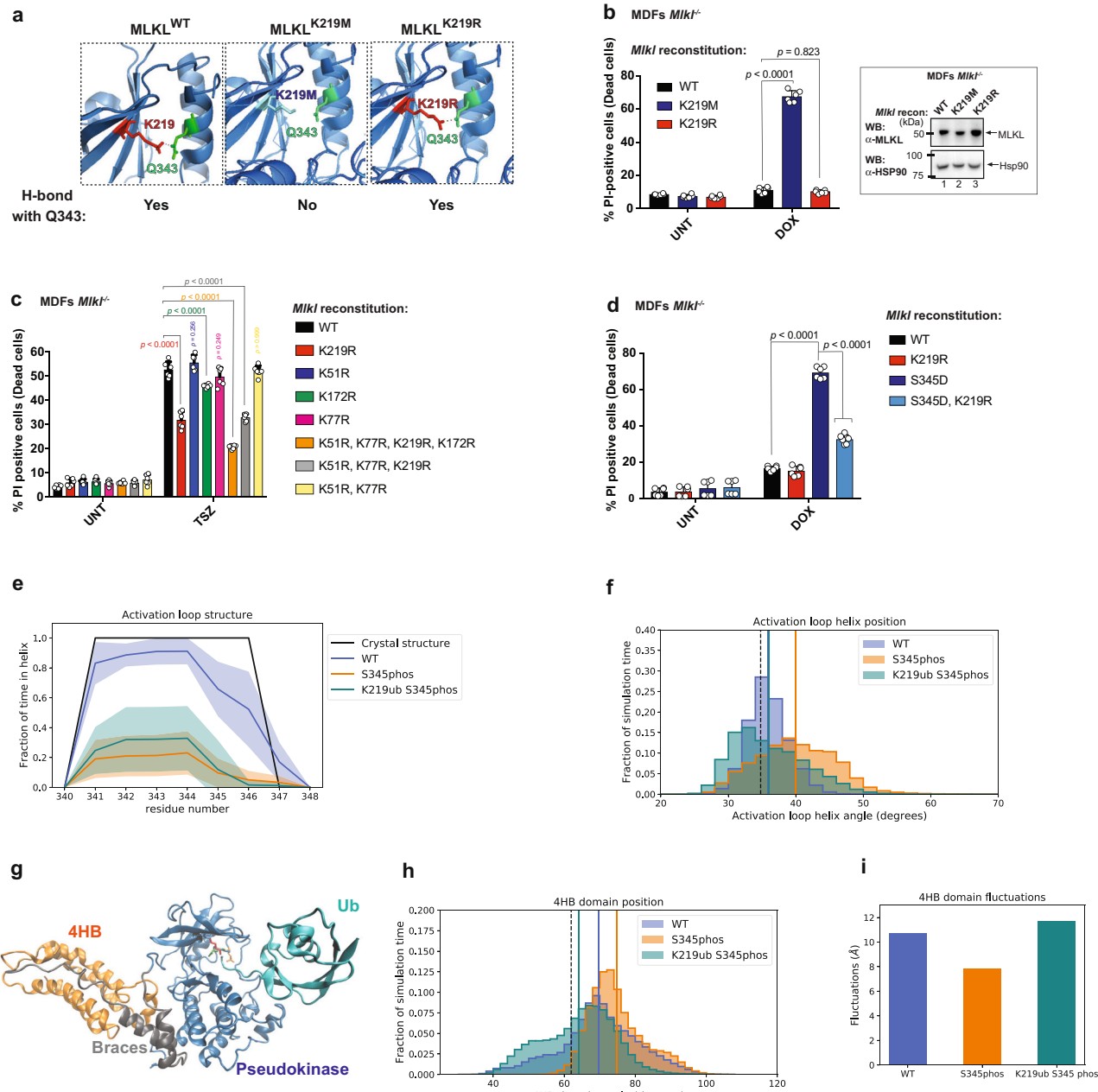

**Fig. 5 Ubiquitylation at K219 contributes to the cytotoxic potential of MLKL. a** Pymol model depicting the formation of a hydrogen bond in MLKL mutants. **b** Quantification of propidium iodide positive (PI⁺) *Mlkl*⁻/⁻ mouse dermal fibroblasts (MDFs) reconstituted with the indicated MLKL mutants and treated for 5 h with doxycycline (DOX, 0.1 μg/ml). The corresponding western blot showing MLKL expression is shown on the right panel. **c** *Mlkl*⁻/⁻ MDF were reconstituted with the indicated MLKL ubiquitin mutants. Cells were pre-treated with DOX (0.1 μg/ml for 3 h) and subsequently treated with TNF (10 ng/ml), SM-164 (100 nM) and z-VAD-FMK (20 μM; TSZ) for 2 h or left untreated. **d** Quantification of PI⁺ *Mlkl*⁻/⁻ MDFs reconstituted with the indicated MLKL mutants and treated for with DOX (0.1 μg/ml for 5 h). In **b**–**d** $n = 6$ wells/group. The results are representative of those from two independent experiments with three technical replicates each. Data are presented as mean ± SD. Statistical analysis shown was calculated by two-way ANOVA with Sidak's multiple comparison test. Source data are provided as a Source data file. **e** Structure of activation loop residues from molecular dynamics (MD) simulations. Shaded area represents standard deviation calculated from independent simulations and the line represents the mean. **f** Histogram of angle between the activation loop helix and the adjacent helix in simulations. The black vertical dashed line represents the angle in the crystal structure. Solid vertical lines represent the average angle from the MD simulations for each construct (the lines for the WT and K219ub P-S345phos simulations are overlapping). **g** Ubiquitylated and phosphorylated MLKL from MD simulations. Shown is the pseudokinase domain (blue), the brace helices (grey), the 4HB domain (orange), the Ub moiety (teal) and residues K219 (red), Q343 (green) and S345 (orange). **h** Histogram of angle between the 4HB domain and the pseudokinase domain in MD simulations. The black vertical dashed line represents the angle in the crystal structure. Solid vertical lines represent the average angle from the MD simulations for each construct. **i** Overall fluctuations of the 4HB domain relative to the pseudokinase domain.

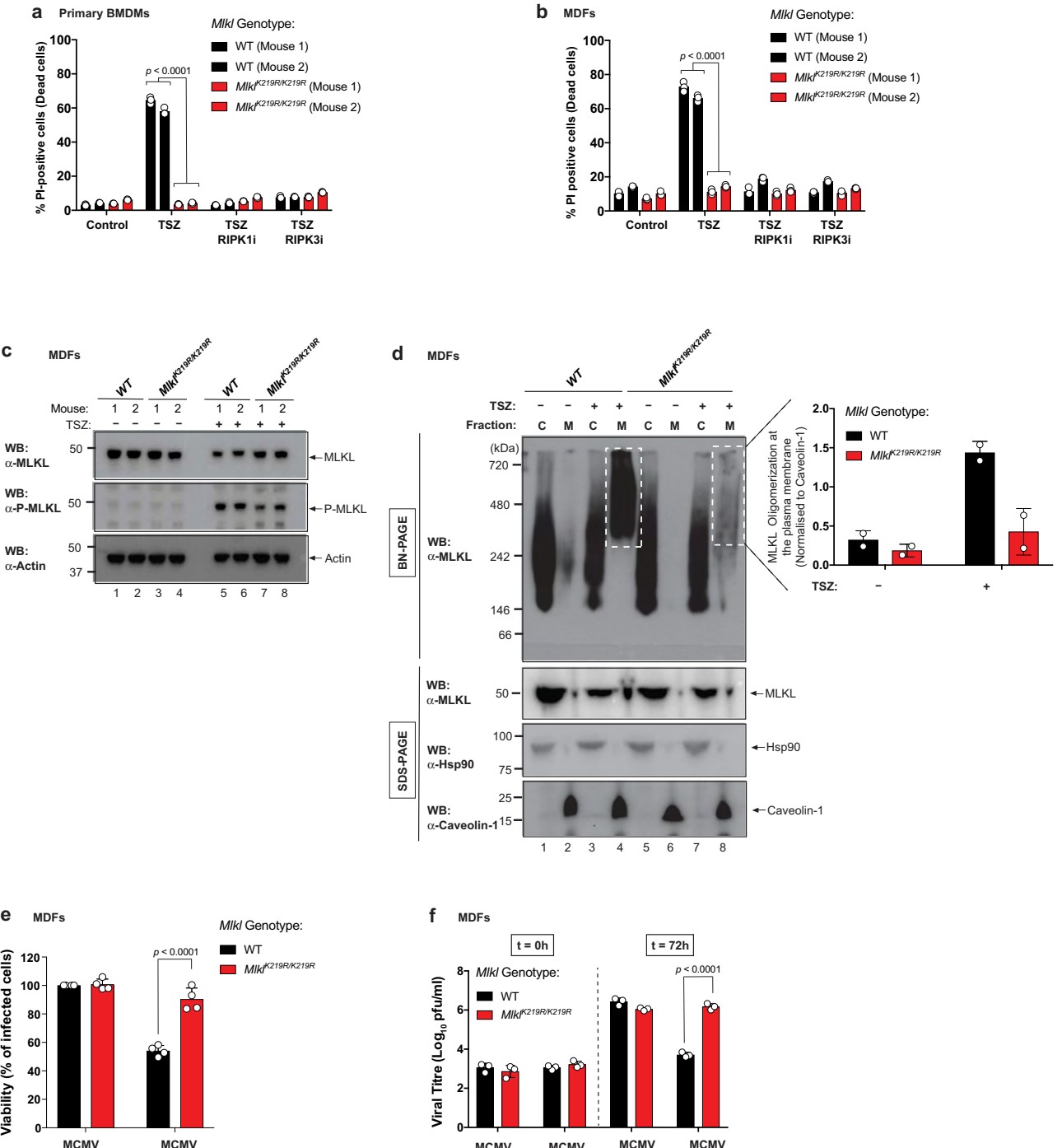

Moreover, like in the murine setting, K230 of hMLKL is ubiquitylated, which is evident as K230 was identified as an Ub acceptor K in a whole ubiquitinome analysis[26], corroborating our findings in murine cells. Furthermore, mutation of K230 renders hMLKL less potent[19], suggesting that, like in the murine setting, ubiquitylation of K230 might contribute to MLKL's killing function. However, due to structural differences between mouse and hMLKL[40], caution should be exercised when extrapolating to the human setting.

Our PLA is consistent with the notion that the conjugation of Ub to MLKL occurs in the cytoplasm. However, since the PLA technique relies on proximity, we cannot rule out the possibility that the obtained signal comes from Ub that is merely proximal,

but not attached to MLKL. Nonetheless, cell fractionation experiments also supported that MLKL ubiquitylation is an early event during necroptosis, and occurs prior to plasma membrane localisation and membrane rupture. At present it remains unclear how ubiquitylation of K219 enhances MLKL polymer formation and cell death. It is possible that ubiquitylation helps to stabilise MLKL's active conformation and consequently, facilitates polymer formation. Indeed, PTMs often control the dynamic toggling between the inactive and active states that are characteristic of protein kinases. Accordingly, Ub modifications frequently occur at sites that are important for kinase regulation[41,42]. For instance, computational simulation of ubiquitylation in key regulatory sites using MD shows that Ub can influence the structure and

**Fig. 6 *Mlkl*$^{K219R/K219R}$ knock-in cells are protected from TNF and MCMV-driven necroptosis. a** Quantification of propidium iodide positive (PI$^+$) primary bone marrow-derived macrophages (BMDMs) derived from two *Mlkl*$^{WT/WT}$ and *Mlkl*$^{K219R/K219R}$ mice. Cell death was measured following treatment with TNF (10 ng/ml), SM-164 (100 nM) and z-VAD-FMK (20 µM; TSZ) for 4 h in presence or absence of RIPK1 inhibitor (RIPK1i GSK'963, 100 nM) or RIPK3 inhibitor (RIPK3i GSK'843 2 µM). **b** Quantification of PI$^+$ mouse dermal fibroblasts (MDFs) derived from two *Mlkl*$^{WT/WT}$ and *Mlkl*$^{K219R/K219R}$ mice following treatment as in **a**. In **a**, **b**, $n = 3$ wells/group. The results are representative of those from two independent experiments with three technical replicates each. Data are presented as mean ± SD. Statistical analysis shown was calculated by two-way ANOVA with Sidak's multiple comparison test. Data from mice with same genotype was pooled together for the statistical analysis in **a** and **b**. **c** Western blot analysis of MLKL expression in MDFs from two pairs of *Mlkl*$^{WT/WT}$ and *Mlkl*$^{K219R/K219R}$ mice, following treatment with TSZ for 2.5 h or left untreated. **d** MLKL oligomerisation analysis. Blue native polyacrylamide gel electrophoresis (BN-PAGE) of the cytoplasmic (C) and membrane (M) fractions of MDFs from *Mlkl*$^{WT/WT}$ and *Mlkl*$^{K219R/K219R}$ mice, following treatment with TSZ for 2.5 h or left untreated. Shown on the right is the quantification of MLKL oligomers at the plasma membrane from two independent experiments. **e** Cell viability assay of *Mlkl*$^{WT/WT}$ and *Mlkl*$^{K219R/K219R}$ MDFs infected with wild-type (WT) and M45 mutant RHIM (M45mutRHIM) murine cytomegalovirus (MCMV$^{WT}$ and MCMV$^{M45mutRHIM}$). **f** Measurement of viral growth. *Mlkl*$^{WT/WT}$ and *Mlkl*$^{K219R/K219R}$ MDFs were infected with the respective virus and the viral titre was determined by plaque assay 72 h post infection. In **e** and **f**, $n = 4$ and 3 wells/group, respectively. The results are representative of those from three independent experiments with three to four technical replicates each. Data are presented as mean ± SD. Statistical analysis shown was calculated by two-way ANOVA with Sidak's multiple comparison test. Source data are provided as a Source data file.

dynamics of ZAP-70 in a site-dependent manner, ultimately affecting its activation state[43]. These observations strengthen the hypothesis that Ub could be directly involved in the molecular switch mechanism of MLKL activation by altering the protein's conformation or flexibility in a way that favours polymer formation and thus, kills more efficiently. Although ubiquitylation can affect protein function directly, the most common mode of regulation by Ub conjugation involves specific Ub receptors. This class of protein recognises and translates the Ub message into cellular phenotypes. Ub receptors carry specialised Ub-binding domains through which they interact with Ub-modified protein. Although Ub conjugation of active MLKL can potentially enhance its cytotoxic activity, it is likely that Ub-binding proteins also contribute to the transport or assembly of MLKL polymers. We would like to speculate that Ub conjugation of MLKL might provide cells with a mechanism of regulating MLKL's pro-necroptotic potency. Therefore, Ub receptors may help to traffic MLKL to the plasma membrane, or control its localization and accumulation, thereby facilitating lytic cell death in response to necroptotic conditions[25,44]. Alternatively, ubiquitylation might enable the recruitment of known MLKL modulators, like TAM kinases, which reportedly phosphorylate MLKL at the plasma membrane to promote necroptosis[45].

Recent studies indicate that cells can tolerate significant amounts of P-MLKL. However, little is known about how such cells establish a buffered threshold of active MLKL. Our observations are consistent with a model, in which the conjugation of Ub may also serve as potential 'brake' to sustain survival of the cell when MLKL activation is limited or reversed. Under such sub-lethal conditions, Ub receptors may help to traffic ubiquitylated MLKL into extracellular vesicles or lysosomes for safe disposal. In this respect it is interesting to note that the ESCRT machinery, which contains a plethora of Ub receptors, acts downstream of MLKL to regulate necroptotic cell death and inflammation[46,47]. Future work will investigate the potential cross-talk between MLKL ubiquitylation and ESCRT-mediated regulation of necroptosis.

While MLKL is robustly ubiquitylated during necroptosis, numerous candidate-based and unbiased approaches were unsuccessful in identifying E3 ligase(s) or Ub receptors that contribute to MLKL-mediated cell death. This may be due to redundancy and/or the involvement of multiple E3 ligases and Ub receptors. Furthermore, at least in MDFs, we could not find any role for Peli-1, an E3 ligase that reportedly influences MLKL-mediated necroptosis[48,49] (Supplementary Fig. 8a–c). Future work will identify components of the Ub-signalling system that influence MLKL-mediated necroptotic cell death and its consequences.

In conclusion, our data indicate that the activity of MLKL is regulated through the combined effects of phosphorylation and ubiquitylation. These modifications are likely to fine-tune MLKL activation to enable a rapid response to diverse danger signals. In light of our findings, it is tempting to speculate that the magnitude and timing of the necroptotic response is influenced by the Ub-signalling system, such as E3 ligases, DUBs and Ub receptors. Given that aberrant lytic cell death can contribute to cytokine release syndrome, tissue damage and illness, a better understanding of how the Ub-signalling system influences MLKL activation and plasma membrane permeabilization might lead to the development of new therapeutic strategies to modulate necroptosis.

## Methods

**Reagents, chemicals and antibodies**. The following reagents were used: complete mini EDTA free protease inhibitor cocktail tablets (Roche), PhosStop EasyPack (Roche), propidium iodide (PI, Sigma-Aldrich), digitonin 5% (Thermo Fisher Scientific), doxycycline (Clontech), Duolink in situ detection reagents green (Sigma-Aldrich), glutathione agarose beads (GE Healthcare), Pierce protein A/G plus agarose (ThermoFisher), Pierce streptavidin agarose (Thermo Fisher Scientific), K63-linkage ubiquitin interacting motif (K63 UIM) and K63-mutant UIM Ctrol (made in house), Met1-linkage-specific Ub-binder (M1-SUB) and GST Ctrol (Made in house), PR-619 (LifeSensors), PTMScan ubiquitin remnant motif (K-ε-GG, Cell signalling), puromycin (Invivogen), recombinant M-CSF (Peprotech), RIPK1 inhibitor GlaxoSmithKline GSK'963 (gift from GSK and Stratech Scientific), RIPK3 inhibitor GlaxoSmithKline GSK'843 (gift from GSK and Stratech Scientific), sequencing grade modified trypsin (Promega), SMAC mimetic (SM-164, kind gift from W. Shaomeng), TMT10plex isobaric label reagent set, 1 × 0.8 mg (Thermo-Fisher), TNFα (Recombinant human and mouse, Enzo), TRAIL (kind gift from H. Walczak), USP21 (kind gift from D. Komander), z-VAD-fmk (Enzo), ASTX660 (Gift from Astex Pharmaceuticals), emricasan (Medkoo). MCMV$^{WT}$ and MCMV$^{M45mutRHIM}$ virus (made in house by James Upton). A list of primers used in this study are supplied as a Supplementary Table 1.

The following plasmids were used: TUBE_Ubqx4_pGEX6P1 (kind gift from M. Gyrd-Hansen), pET104-DEST-UIM3x-(RAP80) WT and mutant (kind gift from N. Mailand), GST-M1-SUB (NEMO UBAN)_pGEX6P1 (kind gift from M. Gyrd-Hansen), pBluescript KS (Stratagene), pTRIPZ (Open Biosystem), psPAX2 (Addgene), pMD2.G (Addgene) and pBABE SV40 (large T antigen Δ89-97-missing Bub1 (kind gift from P. Jat), pLC-GFP (kind gift from B.C. Bornhauser).

The following antibodies were used: anti-actin (Santa Cruz Biotech, sc-1615, 1:2000), anti-caveolin-1 (Santa Cruz Biotech, sc-894, 1:1000), anti-Hsp90 (Santa Cruz Biotech, sc-7947, 1:1000), anti-K48-specific ubiquitin (Millipore, 05-1307, 1:1000), anti-K63-specific Ub (Millipore, 05-1313, 1:1000), anti-M1-specific Ub (gift from Genentech under MTA, 1F11, 1:1000), anti-MLKL (Millipore, MABC604, 1:1000 for immunoblot and 1:25 for PLA), anti-phospho-MLKL S345 (Abcam, ab196436, 1:1000), anti-phospho-MLKL S357/T358 (Abcam, ab187091, 1:1000), anti-RIPK1 (BD, 610459, 1:1000), anti-RIPK3 (Novus Biologicals, NBP1-77299, 1:1000), anti-ubiquitin (Dako, Z0458, 1:1000) and anti-ubiquitin (Novus Biologicals, NB300-129,1:25 for PLA). All secondary antibodies are from Jackson Immuno Research (1:10000). Anti-goat IgG (H + L)-HRP (305-035-003), anti-mouse IgG (H + L)-HRP (115-035-003), anti-rabbit IgG (H + L)-HRP (111-035-003), anti-rat IgG (H + L)-HRP (112-035-003), Phalloidin, Alexa Fluor 633 (ThermoFisher Scientific). The following antibodies were used for the spleen analysis: Fixable Viability Dye eFluor 780 (eBioscience 65-0865-14, 1:1000), Alexa

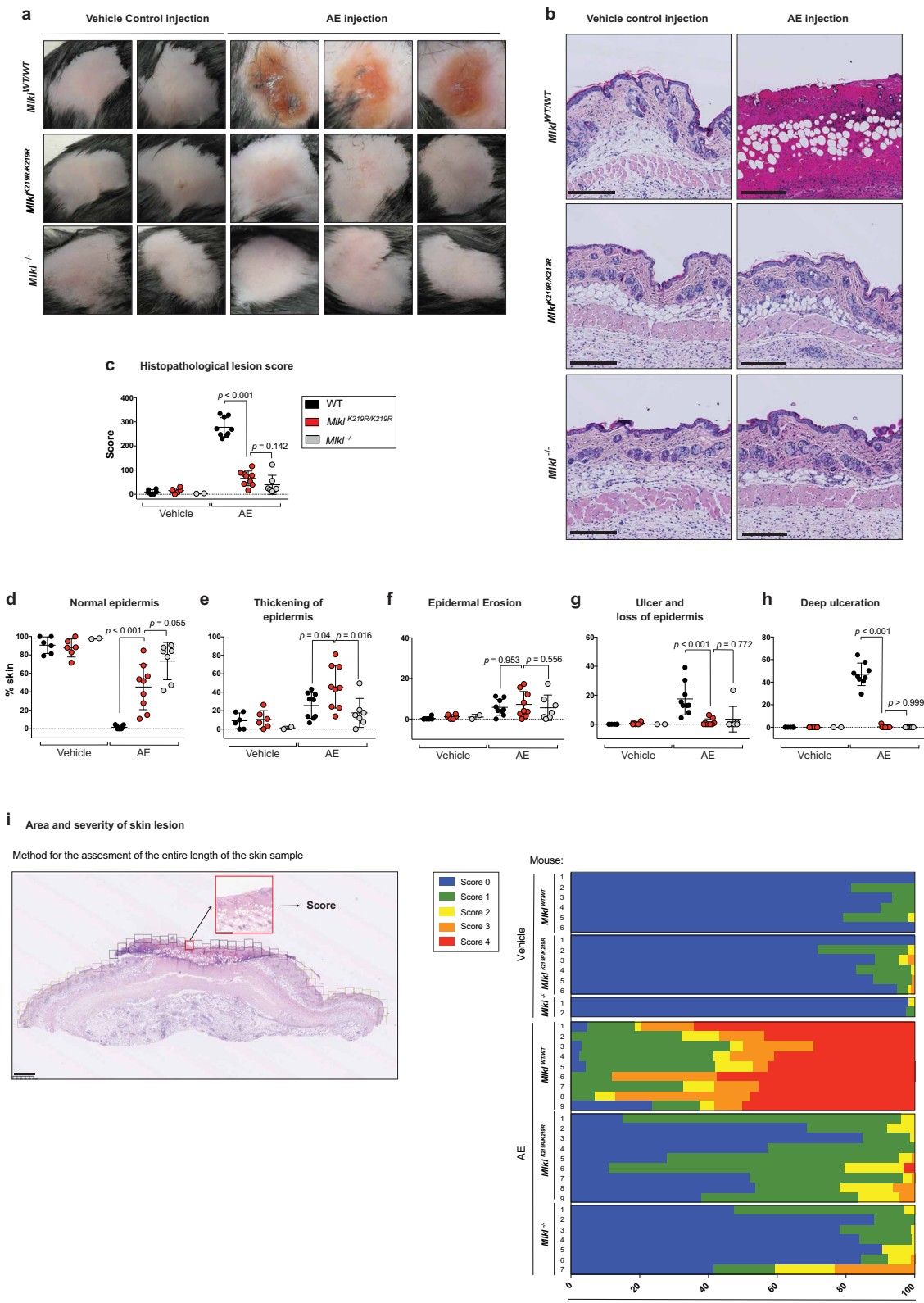

Fluor 700 anti-mouse CD45 clone 30-F11 (BioLegend 103128, 1:200), PE/Cy7 anti-mouse CD3 clone 17A2 (BioLegend 100219, 1:100), PE/Dazzle 594 anti-mouse/human CD45R/B220 clone RA3-6B2 (BioLegend 103257, 1:100), BB515 rat anti-CD11b clone M1/70 (BD 564455, 1:100), BV421 anti-mouse NK-1.1 clone PK136 (BioLegend 108741, 1:100), PE anti-mouse CD11c clone N418 (BioLegend 117307, 1:100), PE/Cy5 anti-mouse F4/80 clone BM8 (BioLegend 123111, 1:100), BV650 anti-mouse Ly-6G clone 1A8 (BioLegend 127641) and BV510 anti-mouse I-A/I-E clone clone M5/114.15.2 (BioLegend 107636, 1:200).

**Mice model generation.** The generation of C57BL/6 $Mlkl^{K219R}$ animals was conducted in the Walter and Eliza Hall Institute of Medical Research (WEHI). Cas9 mRNA together with the ssDNA repair oligo and the sgRNA targeting the region surrounding K219 of the murine $Mlkl$ gene was microinjected into the pronucleus of fertilised oocytes obtained from $C57BL/6$ mice. The injected embryos were transferred to foster mothers and allowed to develop to term. Mutations in the genome of progeny were determined by analysis of genomic DNA and sequencing. The sequence of the ssDNA oligo used as a repair template for the $Mlkl^{K219R}$ can be

**Fig. 7 $Mlkl^{K219R/K219R}$ knock-in mice are protected from necroptosis-driven tissue injury. a** Representative macro images of skin lesions from $Mlkl^{WT/WT}$, $Mlkl^{K219R/K219R}$ and $Mlkl^{-/-}$ mice 72 h after injection with ASTX660/Emricasan (AE) or vehicle control. **b** Representative images of skin sections treated as in **a** and stained with H&E. Scale bars, 250 μM. **c** A final histopathological multivariate lesion score (HLS) of mice treated as described in **a**. The calculation considered the proportional score (%) of given skin lesions within each sample. This was then multiplied by a power score reflecting lesion severity. Final HLS '0' equals normal/regular epidermis and a HLS of 400 equals 100% deep ulceration. $Mlkl^{WT/WT}$ vehicle-treated ($n = 6$), $Mlkl^{K219R/K219R}$ vehicle-treated ($n = 6$) and $Mlkl^{-/-}$ vehicle-treated ($n = 2$). $Mlkl^{WT/WT}$ AE-treated ($n = 9$), $Mlkl^{K219R/K219R}$ AE-treated ($n = 9$) and $Mlkl^{-/-}$ AE-treated ($n = 7$). **d** Percentage contribution of regular epidermis with no changes to stratum corneum, stratum lucidum, stratum granulosum, stratum spinosum, and stratum basale. **e** Percentage contribution of epidermis thickening with (mild to marked) degree of changes to any of the strata: stratum corneum, stratum lucidum, stratum granulosum, stratum spinosum, stratum basale. **f** Percentage contribution of epidermal erosion or partial loss of the epidermis, with the stratum basale left intact. **g** Percentage contribution of ulcer and loss of epidermis, including the stratum basale. **h** Percentage contribution of ulceration with dermal and hypodermal fibrosis and evidence of cell death. Formation of dense collagenous fibrotic tissue. Extensive necrotic area and the presence of keratotic debris. In graphs **c–h**, the score for each mouse is represented by a single dot. The amount of affected skin was scored from 0 to 100% in graphs **d–h**. Data were collected in two independent experiments. Mean ± SD is shown. Statistical significance by two-sided non-parametric Mann–Whitney test. **i** Percentage of individual skin characteristics of mice treated as in **a**. Left panel depicts an example of how each skin section was assessed using the grid. Scale bar, 500 μM. Assessment was performed on the entire length of the skin sample. Two skin sections from each mouse were assessed. Source data are provided as a Source data file.

obtained upon request. $Mlkl^{K219R}$ animals were back-crossed to C57BL/6 mice for at least two generations prior analysis. $Mlkl^{-/-}$ mouse line was a kind gift from Prof. Henning Walczak. Mice were kept according to the UK Home Office regulations. In vivo experiments were conducted according to institutional, national and European animal regulations. Animal protocols were approved by the Ethics Committee of The Institute of Cancer Research.

**Treatment and histological specimen preparation**. All animal procedures were conducted within the guidelines of UK Home Office in accordance with the revised (2013) Animals (Scientific Procedures) Act (ASPA) and the institutional guidelines of The Institute of Cancer Research. The Animal Welfare Ethical Review Body (AWERB) reviewed the protocols within the project license. For the skin injection experiments, mice were injected subcutaneously in the flank region with 100 μl of a solution containing 3 mM ASTX660 and 1 mM emricasan. A solution of 12.5% captisol was used as a vehicle. $Mlkl^{K219R}$ and $Mlkl^{WT}$ and $Mlkl^{-/-}$ mice were injected with vehicle control or with ASTX660 and emricasan. On day 3 post injection, mice were culled and regions of injection were macroscopically assessed. After sacrificing the animals, the fur in the area of lesion and surrounding region was removed post-mortem with hair removal cream. A biopsy of the skin containing the affected area, as well as the lesion boundaries with inclusion of healthy-looking skin (~1.5 cm) was obtained. The skin was fixed in 10% neutral buffered formalin for 24 h, processed (using Sakura tissue processor) and paraffin embedded according to local SOP. A 3 μm thick sections revealing epidermal, dermal and subcutaneous layers were stained with haematoxylin and eosin (H&E) to visualise general tissue morphology. For in vivo study with RIPK1 inhibitor, mice were fed on normal chow (repelleted purina chow 5001, Resesarch Diets) or purina chow 5001 containing RIPK1i (GSK'547 at 0.83 g RIPK1i/Kg chow targeting a dose of ~100 mg/Kg/day, Research Diets)[50] beginning seven days before the experiment was performed, as described above.

**Splenocyte preparation and flow cytometry**. To isolate splenocytes, mouse spleens were mashed through a cell strainer into a Falcon tube using the plunger end of a syringe and collected in 3 ml cold RPMI/10% foetal bovine serum (FBS). After centrifugation, pellets were resuspended in 1 ml 1× red blood cell lysis buffer (eBioscience, 00-4333-57) and incubated for 5 min at room temperature. Cells were washed twice in staining buffer (PBS/5% FBS) and counted. A total of $3 \times 10^6$ splenocytes were stained for each sample. Cells were resuspended in 40 μl of cold staining buffer containing Viability Dye and incubated for 10 min at room temperature. Cells were washed twice and resuspended in 40 μl of staining buffer containing the desired antibodies. Cells were kept on ice for 30 min. After two washes, cells were fixed in 150 μl 2% PFA for 20 min on ice, washed twice and resuspended in staining buffer. Cells were analysed the next day on a BD LSR II flow cytometer.

**Tissue culture**. MDFs, HT-29, L929, NIH3T3 cells (used for MCMV viral titration) and HEK293T (used for lentivirus production) were cultured in Dulbecco's modified Eagle's Medium (DMEM). All media were supplemented with 10% FBS and penicillin and streptomycin, under 10% $CO_2$. Immortalised WT and $Mlkl^{-/-}$ MDFs were a kind gift from John Silke.

**Isolation of primary cells**. To generate BMDMs, the bone marrow from tibias and femurs of $Mlkl^{K219R/K219R}$ and $Mlkl^{WT/WT}$ mice were extracted by placing a needle into the bone and flushing some media with a syringe. After spinning the cells at $160 \times g$ for 5 min, cells were seeded in non-coated petri dishes and cultured for 6 days in DMEM + 10% FBS supplemented with cytokine recombinant murine M-CSF (10 ng/ml). Primary MDF cells were isolated from the tail of littermate adult mice (6–8-week-old) harbouring $Mlkl^{K219R/K219R}$ and $Mlkl^{WT/WT}$ genotypes. To this end, the entire tail of the mouse was cut and placed in a dish containing ethanol for 1 min. A longitudinal incision of 5 mm was made at the top part of the tail and the skin was peeled off using thin tweezers. The tip of the tail was discarded and the skin was placed in a 10 cm dish containing 10 ml DMEM + 10% FBS + Pen/Strep. The skin of the tail was then chopped in small pieces (2 mm) and the pieces were collected in a 15 ml falcon tube. The tube was spin at $160 \times g$ for 3 min at 4 °C. The media was removed and 10 ml of PBS was added. The tube was gently rocked for five times and then centrifuged at $160 \times g$ for 3 min at 4 °C. The PBS was discarded and 3 ml of ice-cold trypsin + EDTA (Sigma) was added and incubated for 45 min at 37 °C. Following incubation, 6 ml of media was added and the tube was shaken vigorously (30×). The resuspended skin was filtered through a 100 μm strainer in a 50 ml falcon tube. The strained media containing the MDFs was placed in a 10 cm dish and incubated at 37 °C, 5% $CO_2$. A couple of days later, MDFs were immortalised with pBABE SV40 (large T antigen Δ89-97-missing Bub1).

**Constructs and transfections**. Murine Mlkl constructs with the following mutations $Mlkl^{S345D}$, $Mlkl^{S345A}$, $Mlkl^{K219M}$, $Mlkl^{K219R}$, $Mlkl^{K172R}$, $Mlkl^{K51R}$, $Mlkl^{K77R}$, $Mlkl^{K51R,K77R}$, $Mlkl^{K51R,K77R,K219R}$, $Mlkl^{K51R,K77R,K219R,K172R}$ and $Mlkl^{K219R,S345D}$ were generated and cloned into pTRIPZ plasmid. Lentiviral supernatants were produced and $Mlkl^{-/-}$ MDFs were infected, as described previously[51]. The information regarding the primers used in this study are supplied as a Supplementary Table. $Mlkl^{N-FLAG}$ lentiviral construct was a kind gift from John Silke. DNA dynamo software was used to verify the mutations of interest.

**Cell death assay**. A total of $2.5 \times 10^4$ BMDMs, $6 \times 10^3$ MDFs, $6 \times 10^3$ L929 cells and $8 \times 10^3$ HT-29 cells were seeded in 96-well plates and 24 h later were treated as indicated for the indicated times. Hoechst (0.5 μg/ml) and PI (1 μg/ml) were added and the % of dead cells was measured, using the Celigo S Cell Imaging Cytometer (Nexcelom Bioscience).

**Generation of CRISPR Cells**. Guide RNA for mRIPK3 and mPeli1 were designed using the following website tools, https://crispr.mit.edu and http://crispor.tefor.net, respectively. sgRNAs were cloned into pLenti-CRISPR-EGFP plasmid (kind gift from B.C. Bornhauser, Addgene #75159). WT MDFs were transfected with the plasmids using Viromer YELLOW transfection reagent (OriGene). Seventy-two hours after transfection GFP-positive clones were FACS sorted and single clones were screened for RIPK3 gene knockout. For Peli-1 deletion, a pool of cells was used instead of single clones. The indel efficiency for Peli-1 pools was calculated with the following website tool, https://www.synthego.com. RIPK3 sgRNA target sequence: GAGGGTTCGGAGTCGTGTTC, Peli-1 sgRNA1: CTTGTGATGCAT CCACGTAA and Peli-1 sgRNA2: TCTCTCCCAAACGGTGATAG.

**MCMV virus infections and determination of viral titres**. Viral titres were determined by plaque assays performed on NIH3T3 fibroblasts. Viral growth in vitro was determined by infection of indicated cell lines at a multiplicity of infection (MOI) of 5 PFU per cell to measure single-step growth kinetics. Viruses were adsorbed for 2 h at 37 °C in a volume of 0.4 ml. Cells and supernatants for quantitation were harvested at indicated times post infection, and frozen at −80 °C. Samples were subjected to one round of freeze/thaw, and virus quantitated by plaque assay on NIH3T3 fibroblasts, as previously described[13].

**Cell viability assay following MCMV infection**. Cell viability assay was performed as previously described[13], using Promega CellTiter-Glo kit at 18–22 h post infection.

**Western blot**. Cells used for expression analyses were washed with ice-cold PBS and lysed using DISC lysis buffer (1 M Tris pH 7.8, 5 M NaCl, 0.5 M EDTA, 40% glycerol and 1% Triton X-100) containing protease inhibitor cocktail and phosphatase inhibitors. Cells were scraped using cell scrapers and lysates were cleared by centrifugation at 4 °C and 20,817 × g for 15 min. Protein concentrations were determined using a BioRad protein Bradford assay in accordance with the manufacturer's instructions. After protein concentration was determined, 6× loading die (3 ml 20% SDS, 4 ml 100% glycerol, 3 ml β-ME and 1 g bromophenol blue) was added to 100 µg of protein diluted in a total volume of 100 µl. Samples were boiled at 100 °C for 10 min and then resolved by NuPAGE Novex 4–12% Bis-Tris 1.0 mm gels in MOPS buffer (ThermoFisher). After transfer onto polyvinylidene difluoride (PVDF), membranes were blocked with 5% BSA and then probed with antibodies as indicated. ImageJ software was used for protein bands quantification.

**MLKL oligomerization assay (non-reducing SDS–PAGE)**. The assay to visualise oligomeric forms of MLKL was performed, as previously described[9,21]. Briefly, cells of a 10-cm dish were washed with ice-cold PBS and lysed in DISC lysis buffer containing protease inhibitor cocktail and phosphatase inhibitors. For non-reducing conditions, β-ME was removed from the 6× loading die and samples were directly loaded into the gel.

**Ubiquitin pulldown**. Following treatment according to the experimental procedures, cells were washed with ice-cold PBS and lysed using DISC lysis buffer supplemented with protease and phosphatase inhibitors, 1 mM DTT, PR-619 (10 µM) and GST-TUBE[52] (50 µg/ml; 50 µg TUBE/mg protein lysate). Cells were scraped using cell scrapers and lysates were cleared by centrifugation at 4 °C and 20,817 × g for 15 min. A total of 20 µl GSH beads were added to each sample and immunoprecipitations were performed o/n. Beads were washed four times in washing buffer (PBS 1% Triton X-100, PBS-Tx) supplemented with 10 µM PR-619 and bound proteins eluted by boiling in 60 µl 1× SDS loading dye. When USP21 DUB digestion was performed, each sample was split in two after the fourth wash. A total of 2 µM of USP21 in 20 µl of 1× DUB digestion buffer (10× buffer: 500 mM Tris (pH 7.5), 500 mM NaCl and 50 mM DTT) was added to the dried beads and digestion was performed for 1 h at 37 °C. For the TUBE experiments were MLKL oligomerization is shown, the TUBE pulldown was performed as described above, but without the DTT supplementation in the lysis buffer. In this case, proteins were eluted in non-reducing loading dye[9] (3 ml 20% SDS, 4 ml 100% glycerol, 3 ml H2O and 1 g bromophenol blue). For K48-specific Ub pulldown, 20 µl protein A/G beads were preincubated with 1.5 µl of α-K48-specific Ub antibody. As a control α-V5 tag antibody was utilised. For K63-linked and M1-linked chains specific ubiquitin tools were utilised. To isolate K63-linked ubiquitin chains, the UIM domains of RAP80 were engineered in tandem (three UIMs)[53]. To isolate M1-linked polyubiquitin chains, a Met1-linkage-specific Ub-binder (M1-SUB) was utilised[54,55]. For the K63-linked Ub pulldown 20 µl streptavidin beads were preincubated with 4 µg K63 UIM recombinant protein or K63 UIM Mutant as a control. For the M1-linkage-specific ubiquitin pulldown 20 µl of GSH beads were mixed with 28.5 µg of M1-SUB or GST control. In all cases, beads were preincubated for 2 h at 4 °C. Following preincubation, equal amounts of clarified lysate were added to the beads and immunoprecipitation was performed o/n.

**Proximity ligation assay**. PLA was performed according to the manufacturer's protocol using the Duolink Detection Kit (Sigma, DUO92014). Cells were examined with a confocal microscope (objective ×40, Zeiss LSM 710).

**Cellular fractionation with Triton X-114**. Cellular fractionation with Triton X-114 was performed, as previously described[23]. Briefly, cells of a 15-cm dish were washed with ice-cold PBS and scraped using 0.5 ml Triton X-114 lysis buffer (20 mM HEPES, pH 7.4, 150 mM NaCl and 2% Triton X-114) supplemented with protease inhibitor cocktail and phosphatase inhibitors. The cell lysate was centrifuged at 15,000 × g for 10 min at 4 °C. The supernatant was harvested as the detergent-soluble fraction. This fraction was warmed at 30 °C for 3 min and subjected to centrifugation at 1500 × g for 5 min at RT. The aqueous layer was collected and recentrifuged at 1500 × g for 5 min at RT to remove the contamination from the detergent-soluble layer and kept as the cytoplasmic fraction (C). The detergent-enriched layer was diluted with basal buffer (20 mM HEPES, pH 7.4 and 150 mM NaCl) and recentrifuged at 1500 × g for 5 min at RT. The washed detergent-enriched layer was diluted with the basal buffer to the same volume as the cytoplasmic fraction and saved as the membrane-enriched fraction (M). Samples fractionated by this method were subjected to downstream processing including ubiquitin pulldown, SDS–PAGE and immunoblotting.

**Blue native PAGE**. For BN-PAGE cells were fractionated into cytoplasmic and membrane fractions, as previously described[11,56]. Briefly, cells are initially permeabilised using a buffer containing 0.025% digitonin (20 mM HEPES, pH 7.5,

100 mM KCl, 2.5 mM MgCl2, 100 mM sucrose, 0.025% digitonin and 2 µM N-ethylmaleimide). Cytosolic and crude membrane fraction were separated via centrifugation at 11,000 × g for 5 min at 4 °C. The cytosolic fraction (top fraction) was further solubilized in permeabilization buffer and digitonin was added to achieve a final concentration of 1% wt/vol. The detergent fraction (bottom fraction) was further permeabilized by adding permeabilization buffer containing 1% wt/vol of digitonin and clarified by centrifugation. Fractions were resolved by 4–16% Bis-Tris native PAGE gel (ThermoFisher) and then transferred onto PVDF for western blot analyses. Following transfer, PVDF membranes were de-stained by washing three times with 40% methanol, 10% acetic acid for 5 min/wash. Prior to immunoblotting, proteins were denatured by soaking the PVDF membrane in 6 M guanidine hydrochloride, 10 mM Tris·HCl pH 7.5 and 5 mM β-ME for 2 h at RT.

**Thermal shift assay**. Thermal shift assays were performed as described previously[8,57–59], using a Rotor-Gene Q PCR (QIAGEN). Recombinant mouse MLKL pseudokinase domain (residues 179-464; WT or K219R) was diluted in 150 mM NaCl, 20 mM Tris pH 8.0, 1 mM DTT to a final concentration of 2.5 µM and assayed in the presence of up to 1 mM ATP in a total reaction volume of 25 µL. SYPRO Orange (Molecular Probes) was used as the fluorescence probe and detected at 550 nm.

**Mass spectrometry sample preparation**. Briefly, MDFs were treated with TNF (10 ng/ml), SM-164 (100 nM) and z-VAD-FMK (20 µM; TSZ) for 2 h, or left untreated and lysed in denaturing lysis buffer (8 M urea, 50 mM Tris pH 8, 50 mM NaCl and 5.5 mM chloroacetamide (ChloroAA)) supplemented with protease inhibitors and phosphatase inhibitor, and the lysate was incubated 20 min on ice. Samples were sonicated and clarified by centrifugation at 20,817 × g for 15 min at 4 °C. Protein concentration was quantified using Bradford protein assay (BioRad) and 20 mg of protein per sample was used as total starting material. Protein was precipitated by adding chloroform and methanol to the lysates. Briefly, lysates were mixed with 4× volumes of 100% methanol and vortexed. Next, 1× volume of chloroform was added to the samples followed by 3× volumes of HPLC H2O. Proteins were precipitated by centrifugation at 7431 × g at RT for 15 min. The pellet was washed by adding 100% methanol and swirling the tube multiple times. A total of three washes with methanol was performed and samples were centrifuged at 7431 × g for 3 min at RT between washes. During the last wash step, the pellet was vortexed for a couple of seconds in methanol to break it into small pieces. The methanol was carefully aspirated and the protein precipitate was re-dissolved in 7 ml denaturation buffer (6 M urea, 2 M thiourea and 50 mM Tris, pH 8). Samples were reduced by adding DTT to a final concentration of 1 mM and incubated 1 h at RT with rotation. ChloroAA (Sigma-Aldrich) was added to a final concentration of 5 mM and samples were incubated for 1 h at RT in the dark with rotation. Proteins were subsequently digested by adding 40 µg of Lysyl Endo-peptidase Mass Spectrometry Grade (Lys-C; Alpha Labs), for 4 h. Subsequently, four volumes (28 ml) of 25 mM TRIS, pH 8 was added to the samples and they were subsequently digested by addition of with 100 µg of trypsin (Roche, 11418475001) and o/n incubation at RT with rotation. Digestion was stopped by adding trifluoroacetic acid (TFA) to a final concentration of 0.5% (1.75 ml of 10% TFA). Samples were incubated for 10 min at RT. Following digestion, peptides were desalted using Sep-Pak Vac tC18 cartridges (Waters). Briefly, cartridges were activated with 9 ml of acetonitrile (ACN, Fisher Scientific). The column was then equilibrated with 10 ml of 0.1% TFA and peptides were loaded into the column. Peptides were washed and desalted using 12 ml of 0.1% TFA and eluted with 5 ml of 50% ACN/0.1%TFA into a 12-ml Duran glass tube (Fisher Scientific). The eluate was frozen in liquid nitrogen and the peptides were dried in a freeze dryer. After drying was finished, dried peptides were collected at the bottom of the tube by centrifugation at 1646 × g 5 min at RT.

**Immunoprecipitation of di-Gly containing peptides**. Immunoprecipitation of di-Gly peptides was performed as previously described[60], albeit with some modifications. Lyophilised peptides from 20 mg of digested proteins were resuspended in 1.3 ml of 1× IAP buffer (50 mM MOPS pH 7.4, 10 mM Na2HPO4 and 50 mM NaCl) by carefully pipetting up and down. The pH was adjusted to 7 by adding 30–90 µl 1 M Tris pH 10 to facilitate peptide resuspension. The resuspended peptides were centrifuged at 20,817 × g for 8 min at 4 °C to remove any insoluble material. The resuspended peptides were incubated with 40 µl of α-di-Gly antibody beads (Cell Signalling) previously washed three times with 750 µl of cold 1× IAP buffer. Immunoprecipitation was performed for 2.5 h at 4 °C with rotation. Following the immunoprecipitation, samples were centrifuged at 2700 × g for 1.5 min at 4 °C and the supernatant was removed. The beads were washed twice with 750 µl cold MilliQ H2O. After the second wash, the beads were resuspended in 100 µl cold MilliQ H2O and placed in a Pierce Spin column placed in a 1.5 ml tube. The columns were spin at 2700 × g, 1.5 min at 4 °C to dry the beads and then they were transferred to a fresh 1.5 ml tube. Di-Gly peptides were eluted by addition of 80 µl of 0.15% TFA followed by centrifugation at 2700 × g for 1.5 min at 4 °C. A second elution was performed by adding another 80 µl of 0.15% TFA to the beads in the column and spin at 2700 × g for 1.5 min at 4 °C into the same tube as the first elution.

**TMT labelling**. The six samples from the di-Gly immunoprecipitation were labelled by TMT10plex Isobaric Mass Tag Labelling Reagents Set (ThermoFisher), according to manufacturer's instructions. The six samples were combined in a new centrifuge tube and the labelled peptide sample was dried in SpeedVac.

**LC–MS/MS analysis for di-Gly immunoprecipitation**. The dried peptide mixture was resuspended in 0.1% $NH_4OH$/100% $H_2O$, and fractionated on an XBridge BEH C18 column (2.1 mm i.d. × 150 mm, Waters) with an initial 5 min loading then linear gradient from 5% ACN/0.1% $NH_4OH$ (pH 10) to 35% $CH_3CN$/0.1% $NH_4OH$ in 30 min, then to 80% $CH_3CN$/0.1% $NH_4OH$ in 5 min and stayed for another 5 min. The flow rate was at 200 μl/min. Fractions were collected ay every 30 s to a 96-well plate by column from retention time at 3–45 min, and then pooled by rows to eight concatenated fractions and dried in SpeedVac. The peptides were reconstituted in 20 μl of 0.1% $FA/H_2O$ and then injected for online LC–MS/MS analysis on the Orbitraip Fusion Lumos hybrid mass spectrometer coupled with an Ultimate 3000 RSLCnano UPLC system (both from ThermoFisher). Samples were first loaded and desalted on a PepMap C18 nano trap (100 μm i.d. × 20 mm, 100 Å, 5 μl) then peptides were separated on a PepMap C18 column (75 μm i.d. × 500 mm, 2 μm) over a linear gradient of 8–32% $CH_3CN$/0.1% FA in 90 min, cycle time at 120 min at a flow rate at 300 nl/min. The MS acquisition used MS3 level quantification with synchronous precursor selection (SPS) with the top speed 3 s cycle time. Briefly, the Orbitrap full MS survey scan was $m/z$ 375–1500 with the resolution 120,000 at $m/z$ 200, with AGC set at 4e5 and 50 ms maximum injection time. Multiply charged ions ($z = 2$–5) with intensity threshold at 1e4 were fragmented in ion trap at 35% collision energy, with AGC at 1e4 and 50 ms maximum injection time, and isolation width at 0.7 Da in quadrupole. The top five MS2 fragment ions were SPS selected with the isolation width at 0.7 Da, and fragmented in HCD at 65% NCE, and detected in the Orbitrap to get the report ions' intensities at a better accuracy.

The raw files were processed with Proteome Discoverer 2.2 (ThermoFisher), using both SequestHT and Mascot (V2.3) search engines to search against the reviewed Uniprot protein database of *Mus musculus* (2018) plus the crap database. The precursor mass tolerance was set at 25 p.p.m. and the fragment ion mass tolerance was set at 0.5 Da. Spectra were searched for fully tryptic peptides with maximum two miss-cleavages. Carbamidomethyl (C) and TMT6plex (peptide N-terminus) were set as static modifications, and the dynamic modifications included deamidation (N, Q), oxidation (M) and TMT6plex (K, GlyGlyK). Peptides were validated by percolator with $q$ value set at 0.05 for the Decoy database search. The search result was filtered by the consensus step, where the protein FDR was set at 0.01 (strict) and 0.05 (relaxed). The TMT10plex reporter ion quantifier used 20 p.p.m. integration tolerance on the most confident centroid peak at the MS3 level. Both unique and razor peptides were used for quantification. Peptides with average reported S/N > 3 were used for protein quantification. Only master proteins were reported.

**LC–MS/MS statistical analysis**. To calculate the fold change in di-Gly peptide abundance, the data were first transformed to log base 2. This transformation normalises the data and allows for analysis using parametric statistical tests. To determine the statistical significance, the normalised abundance of di-Gly peptides were analysed via an unpaired, two-tailed Student's $t$ test using Perseus Software. Statistical significance is indicated via a volcano plot generated with Perseus Software, where the $p$ value and the difference in peptide abundance is represented. For this analysis, the three replicates were taken into consideration.

**Structural predictions of mutations**. To predict the effect of mutating K219, Pymol Software (https://www.pymol.org) was utilised on the resolved structure of mMLKL (PBD: 4BTF)[8].

**Molecular dynamics simulations**. We performed all-atom MD simulations starting from three different MLKL constructs: unmodified MLKL (WT), MLKL phosphorylated at S345 (S345phos), and MLKL monoubiquitylated at K219 and phosphorylated at S345 (K219ub S345phos). All three constructs were built from the crystal structure of mMLKL (PBD: 4BTF)[8]. There were three missing loops and a missing C-terminus in the crystal structure. Modeller was used to build in these missing residues so that all 464 residues of mMLKL were present in the structures used for MD simulations[61]. Missing residues 81–91 were modelled as a helix. We prepared the phosphorylated MLKL structures by modifying the PDB file manually to change the residue name from SER to SEP. We used previously published parameters to simulate the phosphorylated serine in Amber[62,63]. We prepared the ubiquitylated MLKL structure using a crystal structure of diUb, 3NS8 [75], as in refs.[43,64].

All simulations were run with Amber 18 using the Amber ff99sb forcefield[65], and the CUDA version of pmemd in Amber 18 was used to run the simulations on GPUs[66]. All simulations were solvated with TIP3P-FB water such that the edge of the box was at least 16 Å away from the protein before equilibration[67]. Ions were added to neutralise the system: three chloride ions for the WT and K219ub S345phos simulations and one chloride ion for the S345phos simulations. All systems were subject to two rounds of energy minimisation of 1000 steps, where the first 500 steps were steepest descent and the second 500 steps conjugate

gradient in both cases. The systems were then subject to heating from 100 to 300 K in the NVT ensemble (40 ps with harmonic restraints on the protein with a force constant of 10 kcal/mol), and two rounds of equilibration in the NPT ensemble (50 ps with harmonic restraints on the protein with a force constant of 10 kcal/mol and the Berendsen barostat, and 200 ps without restraints using a Monte Carlo barostat). Ten independent production simulations for each construct were then started with new random velocities. Bonds to hydrogen were constrained using the SHAKE algorithm during all simulations. The particle-mesh Ewald procedure was used to handle long-range electrostatic interactions with a non-bonded cut off of 9 Å for the direct space sum. Production simulations were run in the NPT ensemble at 300 K, using Langevin dynamics with a collision frequency of 1.0 ps$^{-1}$, and 1.013 bar, using a Monte Carlo barostat with 100 steps between volume change attempts, with an integration step every 2 fs and coordinates stored every 10 ps. Each production simulation was run for 1 μs, for a total of 10 μs of simulation time for each construct.

We analysed the MD trajectories using the *cpptraj* module of Amber 18 tools[66] and in-house python scripts. All histograms were computed by combining all independent simulations for a given construct (snapshots spaced every 10 ps) and binning over this total ensemble of structures. To determine $p$ values comparing the distribution of various structural metrics for different constructs, we treated the mean value from each of the ten independent simulations as a single data point and used a permutation test with 1,000,000 permutations. Protein structure figures were created using VMD[68].

**Histopathological assessment of murine skin samples**. For the purpose of histopathological sample analysis, skin sections were scanned at ×40 (0.23 μm/pixel), using a NanoZoomer-XR Hamamatsu Photonics (Japan). The analysis of digitised skin samples involved assessment of the vicissitudes in epidermal and dermal region of the skin identifiable on H&E-stained samples.

First, a HLS (multivariate) was used to assess the proportion and severity of histopathological changes. This assessment was performed on the entire length of the skin sample (epidermal and immediate dermal level). In order to increase precision of the assessment, each sample was divided using a grid with equal size of fields of view (FOV) (0.04 mm²/800 μm perimeter; Fig. 7i). Each FOV was assessed for presence of regular epidermis or any pathological changes to regular epidermis and dermis. A final calculation took into account the proportion [%] of given skin lesions within each sample, multiplied by a power score ranging from 0 to 4. Score 0 was associated with regular epidermis with no changes to any of the strata. Score 1 represented thickening of the epidermis represented by any of the strata. Score 2 with epidermal erosion (partial loss of the epidermis), with the stratum basale left intact. Score 3 with ulcer-loss of epidermis, including the stratum basale. Score 4 with ulceration and dermal/hypodermal fibrosis. The final score (HLS) indicates the severity following the formula: $[(0 \times \% \text{ Score } 0) + (1 \times \% \text{ Score } 1) + (2 \times \% \text{ Score } 2) + (3 \times \% \text{ Score } 3) + (4 \times \% \text{ Score } 4)]$. The lowest possible HLS is '0' which is equal to 100% of normal/regular epidermis in the whole sample, and a maximum HLS of '400' equal to 100% of deep ulceration. Supplementary Fig. 5a shows additional detail. NDP.view2 software was used for image visualisation.

**Statistical analysis**. Graphs and statistical analysis were performed using GraphPad Prism V7.0. The statistical analysis performed for each dataset is described in the correspondent figure legend.

**Reporting summary**. Further information on research design is available in the Nature Research Reporting Summary linked to this article.

## Data availability
The mass spectrometry proteomics data (corresponding to Fig. 4a) that support the findings of this study have been deposited in the ProteomeXchange Consortium via the PRIDE partner repository with the dataset identifier PXD018857. All unique biological materials used are available upon reasonable request. Source data are provided with this paper.

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

## Acknowledgements
The authors are indebted to J. Silke, H. Walczak, D. Komander, B.C. Bornhauser, M. Gyrd-Hansen, P. Jat, N. Mailand and S. Wang for reagents. We thank the WEHI MAGEC Facility for the generation of the knock-in mouse. K.A.B. thanks Michael Donnelly for computational support. K.A.B. thanks the MERCURY Consortium for mentoring support. We also thank members of the Meier lab for helpful discussions. Work in the Meier lab is funded by Breast Cancer Now as part of Programme Funding to the Breast Cancer Now Toby Robins Research Centre (CTR-QR14-007) and postgraduate studentships from Cancer Research UK (CRUK; CRM089X). We thank the Breast Cancer Now Toby Robins Research Centre Nina Barough Pathology Core Facility for pathology support. J.W.U. is supported by NIH AI135709, and J.M.M. is funded by NHMRC fellowship (1105754 and 1172929), project (1124735) and IRIISS (9000587) support and the Victorian Government Operational Infrastructure Support scheme. This work was supported by National Science Foundation (https://www.nsf.gov/) award MCB-1852677 to K.A.B. We acknowledge NHS funding to the NIHR Biomedical Research Centre.

## Author contributions
L.R.G., K.A.B., J.W.U. and P.M. designed the research and wrote the paper. L.R.G. performed and analysed experiments in Figs. 1, 2, 3b, d, 4, 5a–d, 6a–d and Supplementary Figs. 1, 2, 3b, c, 4, 8. L.R.G. also generated the figures. T.T, R.N. and W.F. designed and performed experiments in Fig. 7 and Supplementary Fig. 7, and assisted with data analysis and experimental design. R.O.H. designed and performed experiments in Fig. 6e, f. G.L. designed, performed and analysed experiments in Fig. 3a. S.W.J. performed experiment in Fig. 3c, Supplementary Fig. 5b and assisted with cloning. A.A. assisted with the experimental design and the preparation of MDFs and BMDMs. L.Y. run and analysed the mass spec samples in Fig. 4a. N.G. assisted with histology tissue processing. H.K. assisted with CRISPR-mediated knockout in Supplementary Fig. 8. M.P. and J.S.C. assisted with data analysis in Fig. 4a. S.N.Y., C.F., L.-Y.L., I.S.L. and J.M.M. designed, performed and analysed experiment in Supplementary Fig. 3a. J.M.M. also contributed to the manuscript review. A.K. designed the strategy and generated the knock-in mouse. I.R. and P.G. assisted with analysis of the histological tissue sections. K.A.B. designed and performed experiments in Fig. 5e–j and Supplementary Fig. 3d, e. M.S, T.S. and G.W. advised on the in vivo application of ASTX660 compound. J.B., A.M.B. and B.G. developed the chow-containing RIPK1i (GSK′547).

## Competing interests
The authors declare no competing interests.
