## [Peer Review File · Nature Communications]

REVIEWER COMMENTS

Reviewer #1 (Remarks to the Author):

In this report, the authors showed that ubiquitination of MLKL at Lysine 219 plays a key role in promoting MLKL oligomerization and membrane targeting. Using pulldown by TUBE, the authors found MLKL among several necroptosis adaptors to be modified by K63-linked polyubiquitination. Inhibition of RIPK1, RIPK3 kinase activities prevented MLKL ubiquitination. Using mass spectrometry and mutagenesis, the authors identified K219 as a key acceptor site for MLKL ubiquitination. The K219R mutant blocked MLKL ubiquitination but not MLKL phosphorylation at S345, indicating that MLKL phosphorylation precedes ubiquitination at K219. Molecular dynamics and structural simulation suggest that K219 forms hydrogen bonds with Q343 in the pseudokinase domain to restrict MLKL in the inactive conformation. K219 ubiquitination relieves this interaction and thus allows MLKL to switch to its active open conformation. In contrast to K219R, the cancer-associated K219M mutation disrupts this interaction and thus results in spontaneous necroptosis. Finally, the authors generated a knockin mouse expressing MLKL-K219R. Cells from these mice were resistant to TSZ-induced necroptosis and infection-induced cell death by MCMV-M45 mutant virus. The MLKL-K219R mice were resistant to Smac mimetic and caspase inhibitor-induced skin inflammation model.

Overall, these results revealed an interesting and novel regulatory mechanism of MLKL. The results presented are generally robust and rigorous.

Minor comment:

Fig. 3b was meant to show that p-MLKL precedes MLKL ubiquitination and membrane targeting. However, the increase in ubiquitinated MLKL between $t=2$ and $t=2.5$ hours was not visually obvious. One solution will be to quantify the results in this figure.

Reviewer #2 (Remarks to the Author):

The manuscript NCOMMS-20-32300 entitled "Ubiquitylation of MLKL at lysine 219 positively regulates necroptosis-induced tissue injury and pathogen clearance" by Garcia, Meier and colleagues describes in great detail the effects of K219-UbK63 on mouse MLKL function. The authors observe that MLKL is Ub (ubiquitylated) upon necroptosis stimulation, that phosphor-MLKL is also Ub, and that Ub occurs early in the activation process while MLKL is cytosolic. The authors find MLKL-Ub translocated at the membrane and show that higher order MLKL species are phosphorylated and Ub. They then map Ub sites by unbiased mass spectrometry to 4 lysines and demonstrate that the one that matters to functional regulation is K219. This residue is buried in the N-terminal lobe and is a coordination point to Q343 located on the unique helix found in the activation loop of the pseudokinase domain. Phosphorylation of the nearby S345 by RIPK3 is known to trigger MLKL activation but this mechanism remains poorly understood. The authors use MD simulations to explain the dynamics in MLKL conformational rearrangements in apo, S345-P activated, and K219-Ub S345-P proposing that K219-Ub may act to stabilize the active conformation with the 4HB exposed. The authors then show that reconstitution of *mlkl*^{-/-} cells with K218R MLKL severely impairs necroptotic response. Further they create healthy knockin mice with K219R mutations in both alleles and show that cells from these mice are resistant to necroptosis stimulation similar to *mlkl*^{-/-} derived counterparts and that they are resistant to viral induced necroptosis similar to the *mlkl*^{-/-} cells. Finally, the authors use a skin injury model to show that K218R KI mice are very similarly resistant to a subcutaneous treatment with IAP/caspase inhibitors which induced ulceration of the skin.

The experiments are well executed, and the manuscript is easy to read containing a wealth of novel information about the regulation of mouse MLKL in necroptosis.

I have a few comments to help tighten up some of the concepts related to MLKL activation.

1. It is clear that human and mouse MLKL are differentially regulated: active human MLKL may tetramerize while active mouse MLKL is a trimer; the 4HB is more tightly regulated in the human MLKL than in the mouse; the authors have suggested that Ub of K230 may regulate human MLKL although the data presented is merely tantalizing and not rigorously described. It would be good for the authors to make these distinctions before a similar analysis is presented for human MLKL. The authors could even consider removing the sparse data on human MLKL as an option.
2. While the MD simulations suggest opening of mouse MLKL it is puzzling how a poorly accessible K219 may be targeted by an E3 ligase. I want to draw a parallel with the TAM kinases which are thought to phosphorylate a buried Tyr residue of MLKL. Both of these studies suggest that the buried sites may be accessible when the protein is potentially unfolded. Of course, we know nothing about states where MLKL is unfolded but some have claimed that heat shock proteins HSP70 and HSP90 regulate MLKL clients. The authors should cite Mol Cell. 2019 Aug 8;75(3):457-468.e4. doi: 10.1016/j.molcel.2019.05.022 (TAM kinase story); and well as Sci Rep. 2019 Nov 14;9(1):16853. doi: 10.1038/s41598-019-53078-5 (MD simulation of apo and phosphorylated MLKL) and some of the articles that suggest HSP regulation of necrosome components.
3. If the authors tested the pattern of K51, K77, K172, K219 ubiquitylation they could include this data. It would be interesting to see if mutation in one site blocks Ub in others (for instance K219R affecting other sites). This could inform on the possible initial modification.
4. The results with reconstituted K219R mutant in mlkl^{-/-} MDFs vs MDFs from KI K219R/K219R mice show that the reconstitution restores half of the necroptotic response compared to WT but that KI does not restore any necroptosis. Could the authors comment on these discrepancies and is it simply due to different levels in the two different cell line pairs?

Reviewer #3 (Remarks to the Author):

Nature Communications manuscript NCOMMS-20-32300

"Ubiquitylation of MLKL at lysine 219 positively regulates necroptosis-induced tissue injury and pathogen clearance", by Ramos Garcia and colleagues

In this manuscript, the authors report ubiquitylation of MLKL during necroptosis induction by TSZ and TRAILSZ, but not during apoptosis in response to TS. They position the ubiquitylation event downstream of RIPK1/3, subsequent to the phosphorylation of MLKL on S345 by RIPK3, but prior to the plasma membrane localization of MLKL. Using linkage specific ubiquitin pulldowns, they identified K63-linkages as the major constituent of the ubiquitin chains conjugated to MLKL in response to TSZ. Through MS analysis, they then identified 4 lysine residues of MLKL being ubiquitylated following TSZ, and reconstitution of Mlkl^{-/-} cells with K>R mutants revealed a major role of K219 for MLKL cytotoxicity under this challenge. To evaluate the physiological relevance of these findings, the authors generated transgenic MlklK219R mice by CRISPR/Cas9, and report viability of the mice. However, they show that cells derived from these animals are protected from the necroptotic trigger TSZ, and that the mutation alters MLKL oligomerization at the plasma membrane. Finally, and in accordance with their findings, they show that MlklK219R/K219R MDFs succumb less to MCMV M45mutRHIM infection, and that MlklK219R/K219R mice are protected from necroptosis-induced injury caused by subcutaneous injection of IAP antagonist + caspase inhibitor.

Comments:

This is a clear and very well written manuscript presenting novel and high quality results on the importance of K219 ubiquitylation for MLKL cytotoxicity. The conclusions drawn by the authors are supported by solid data. The study therefore provides important new insights on the regulation of

TNF-induced necroptosis, and most probably on MLKL-induced necroptosis in general. I however still have some concerns that I believe should be addressed by the authors.

Comments

1)The authors should show the effect of the K219R mutation on the K63-ubiquitylation of MLKL in response to TSZ. Indeed, Figures 1-3 are dedicated to revealing ubiquitylation of MLKL during necroptosis, but there is no data showing that ubiquitylation is reduced in the K219R mutant.

2)The authors should not only discuss but also experimentally evaluate the importance of MLKL ubiquitylation on K230 in human cells. This is required to demonstrate the conserved aspect of this regulatory mechanism.

3)A smear of S345A MLKL is still pulled down by TUBE (ub) upon TSZ stimulation (Fig. 2d). This seems to be contradictory to the statement of the authors that phosphorylation is a prerequisite for ubiquitylation. The authors should perform a DUB treatment to exclude the ubiquitylation nature of the smear. Also, how does the authors explain the increased binding of non-modified S345A MLKL to the TUBE upon TSZ in Fig. 2d (not observed in RIPK3i conditions – Fig. 2b-c)? Would it mean that non-ubiquitylated MLKL is bound to other ubiquitylated proteins in response to TSZ? This would give weight to my concerns regarding the PLA results (see below).

4)It would be of interest to know the identity of the E3 ligase conjugating MLKL with ubiquitin chains. While performing an unbiased screen to identified this E3 is certainly out of the scope of this study, the authors could still test whether Peli1 ubiquitylate MLKL. To my knowledge, Peli1 is indeed the only pro-necroptosis E3 ligase identified so far (PMID:29078411).

5)It is somehow disappointing that the only in vivo data evaluating the importance of K219 ubiquitylation of MLKL are obtained in an artificial and rather irrelevant model (subcutaneous injection of AE). It would have been much more interesting the cross the mice with a genetic model of disease driven by necroptosis. Since the authors already used MCMV M45mutRHIM in vitro, they could at least additionally perform in vivo infection experiments to further demonstrate the in vivo importance of K219 ubiquitylation of MLKL.

Additional comments

6)The authors should provide an explanation for the presence (and sometimes absence) of non-ubiquitylated MLKL in the different ubiquitin pulldowns. Also, the authors write that MLKL is massively ubiquitylated, but this statement may need to be tuned down as it rather looks like only a minor fraction of MLKL is ubiquitylated (for example Fig. 1C).

7)As MLKL is most probably in complex with other ubiquitylated proteins (such as RIPK1 or RIPK3), the PLA results appear poorly informative. Indeed, they could just reflect non-ubiquitylated MLKL in complex with other ubiquitylated proteins. This possibility should at least be mentioned in the discussion.

8)Fig. 2D, the arrow for P-MLKL does not seem to be at the correct place.

9)How do the authors explain the signal corresponding to non-modified MLKL in Fig. 3C? Non-specific binding only revealed under TSZ?

10)Fig. 7B-C, the staining is very light and the histology not very apparent.

11)Fig. 6D the authors should show MLKL levels by SDS-PAGE in the lysate of each fraction.

12)Line 101-110: The authors claim that ubiquitylation would precede oligomerization. However, the results are only correlative. In order to demonstrate this point, the authors should evaluate the ubiquitylation status of an MLKL mutant that cannot oligomerize.

Reviewer #1 (Remarks to the Author):

In this report, the authors showed that ubiquitination of MLKL at Lysine 219 plays a key role in promoting MLKL oligomerization and membrane targeting. Using pulldown by TUBE, the authors found MLKL among several necroptosis adaptors to be modified by K63-linked polyubiquitination. Inhibition of RIPK1, RIPK3 kinase activities prevented MLKL ubiquitination. Using mass spectrometry and mutagenesis, the authors identified K219 as a key acceptor site for MLKL ubiquitination. The K219R mutant blocked MLKL ubiquitination but not MLKL phosphorylation at S345, indicating that MLKL phosphorylation precedes ubiquitination at K219. Molecular dynamics and structural simulation suggest that K219 forms hydrogen bonds with Q343 in the pseudokinase domain to restrict MLKL in the inactive conformation. K219 ubiquitination relieves this interaction and thus allows MLKL to switch to its active open conformation. In contrast to K219R, the cancer-associated K219M mutation disrupts this interaction and thus results in spontaneous necroptosis. Finally, the authors generated a knockin mouse expressing MLKL-K219R. Cells from these mice were resistant to TSZ-induced necroptosis and infection-induced cell death by MCMV-M45 mutant virus. The MLKL-K219R mice were resistant to Smac mimetic and caspase inhibitor-induced skin inflammation model.

Overall, these results revealed an interesting and novel regulatory mechanism of MLKL. The results presented are generally robust and rigorous.

Minor comment:

Fig. 3b was meant to show that p-MLKL precedes MLKL ubiquitination and membrane targeting. However, the increase in ubiquitinated MLKL between t=2 and t=2.5 hours was not visually obvious. One solution will be to quantify the results in this figure.

We have followed the reviewer's suggestion and have quantified the increase in ubiquitylation of MLKL (P-MLKL and total MLKL) at 2 and 2.5 h following TSZ treatment. As shown in **new Figure 3b** (top panel and middle panel), we found that ubiquitylation of both P-MLKL and MLKL at the plasma membrane increases over time.

The aim of this experiment was to evaluate whether plasma membrane-localised MLKL carries both Ubiquitin- and Phospho-modifications. Our data indeed support this view. Of note, this experiment did not attempt to address the epistatic relationship of phosphorylation and ubiquitylation. The epistatic relationship between the two posttranslational modifications was addressed in **Fig. 2D**, where we demonstrate that preventing phosphorylation at Ser345 impairs ubiquitylation of MLKL. The data from Fig. 2D suggest that phosphorylation of MLKL is a prerequisite for MLKL ubiquitylation, at least for the most abundant, lower molecular Ub~MLKL forms.

We have amended our ms to include the quantification. See **new Figure 3b**, as shown below.

Reviewer #2 (Remarks to the Author):

The manuscript NCOMMS-20-32300 entitled “Ubiquitylation of MLKL at lysine 219 positively regulates necroptosis-induced tissue injury and pathogen clearance” by Garcia, Meier and colleagues describes in great detail the effects of K219-UbK63 on mouse MLKL function. The authors observe that MLKL is Ub (ubiquitylated) upon necroptosis stimulation, that phosphor-MLKL is also Ub, and that Ub occurs early in the activation process while MLKL is cytosolic. The authors find MLKL-Ub translocated at the membrane and show that higher order MLKL species are phosphorylated and Ub. They then map Ub sites by unbiased mass spectrometry to 4 lysines and demonstrate that the one that matters to functional regulation is K219. This residue is buried in the N-terminal lobe and is a coordination point to Q343 located on the unique helix found in the activation loop of the pseudokinase domain. Phosphorylation of the nearby S345 by RIPK3 is known to trigger MLKL activation but this mechanism remains poorly understood. The authors use MD simulations to explain the dynamics in MLKL conformational rearrangements in apo, S345-P activated, and K219-Ub S345-P proposing that K219-Ub may act to stabilize the active conformation with the 4HB exposed. The authors then show that reconstitution of *mlkl*^{-/-} cells with K218R MLKL severely impairs necroptotic response. Further they create healthy knockin mice with K219R mutations in both alleles and show that cells from these mice are resistant to necroptosis stimulation similar to *mlkl*^{-/-} derived counterparts and that they are resistant to viral induced necroptosis similar to the *mlkl*^{-/-} cells. Finally, the authors use a skin injury model to show that K218R KI mice are very similarly resistant to a subcutaneous treatment with IAP/caspase inhibitors which induced ulceration of the skin.

The experiments are well executed, and the manuscript is easy to read containing a wealth of novel information about the regulation of mouse MLKL in necroptosis.

I have a few comments to help tighten up some of the concepts related to MLKL activation.

1. It is clear that human and mouse MLKL are differentially regulated: active human MLKL may tetramerize while active mouse MLKL is a trimer; the 4HB is more tightly regulated in the human MLKL than in the mouse; the authors have suggested that Ub of K230 may regulate human MLKL although the data presented is merely tantalizing and not rigorously described. It would be good for the authors to make these distinctions before a similar analysis is presented for human MLKL. The authors could even consider removing the sparse data on human MLKL as an option.

We entirely agree with reviewer 2. This point was also raised by Reviewer 3.

Instead of deleting the data on human MLKL, we followed the suggestions by Reviewer 3 and expanded our manuscript by highlighting that K230 of human MLKL is also ubiquitylated, and that K230M mutation renders human MLKL less active. We thought that this information will be of interest. However, due to the structural differences between mouse and human MLKL, it is difficult to extrapolate to the human setting.

Correlative data from the human setting:

- 1) We find that human MLKL, like its mouse counterpart, is ubiquitylated during necroptosis (Figure 1c)
- 2) Mass spec-based proteomic approaches identified K230 of human MLKL as being ubiquitylated (www.phosphosite.com, [1]). Of note, K230 of human MLKL corresponds to K219 of mouse MLKL.
- 3) Mutation of K230 has previously been found to suppress the cytotoxic activity of human MLKL [2].

These data are consistent with the notion that human MLKL is ubiquitylated at K230 during necroptosis and that K230 might contribute to the cytotoxic potential of human MLKL. This point is discussed in the

Discussion section (page 15, lines 365-373) where we now state that the extrapolation to the human setting should be made with caution due to the structural differences between mouse and human.

2. While the MD simulations suggest opening of mouse MLKL it is puzzling how a poorly accessible K219 may be targeted by an E3 ligase. I want to draw a parallel with the TAM kinases which are thought to phosphorylate a buried Tyr residue of MLKL. Both of these studies suggest that the buried sites may be accessible when the protein is potentially unfolded. Of course, we know nothing about states where MLKL is unfolded but some have claimed that heat shock proteins HSP70 and HSP90 regulate MLKL clients. The authors should cite Mol Cell. 2019 Aug 8;75(3):457-468.e4. doi: 10.1016/j.molcel.2019.05.022 (TAM kinase story); and well as Sci Rep. 2019 Nov 14;9(1):16853. doi: 10.1038/s41598-019-53078-5 (MD simulation of apo and phosphorylated MLKL) and some of the articles that suggest HSP regulation of necrosome components.

We thank the reviewer for highlighting these important points. We agree that MLKL is a client pseudokinase of HSPs and that this interaction might be important for MLKL regulation. We have expanded our text to better reflect this issue. The corresponding citations have now been incorporated in the Discussion section of the revised manuscript (page 15, lines 362-365).

3. If the authors tested the pattern of K51, K77, K172, K219 ubiquitylation they could include this data. It would be interesting to see if mutation in one site blocks Ub in others (for instance K219R affecting other sites). This could inform on the possible initial modification.

We have followed the reviewer's suggestion and have included our data on the ubiquitylation pattern of *MIK*^{K219R} in cells from knock-in animals. *MIK*^{K219R} showed a reduction of the Ub smearing pattern (**Supplementary Fig. 4d**). Thus, mutation of K219 alone can already reduce K63-linked MLKL ubiquitylation.

Further, we find that *MIK*^{K51R,K77R,K172R,K219R} is less ubiquitylated than WT MLKL (**Supplementary Fig. 5d**). Under this reconstitution setting, none of the Lys mutations completely abrogate MLKL ubiquitylation. This may be due to the fact that i) there are additional Ub-acceptor Lys residues in MLKL, or ii) upon mutation of the respective Ub acceptor Lys, the E3 ligase transfers Ub to an alternative Lys nearby. The latter is frequently observed upon K>R mutations [3, 4].

Reviewer 2 suggested to study the epistatic relationships of the respective ubiquitylation sites. This is an interesting idea, although one that it is not easily addressed with the available tools. While it is known that phosphorylation sites can display epistatic relationships, to the best of our knowledge such epistatic relationships have not been described for ubiquitylation sites. To test this, we would have to conduct mass spec-based experiments using absolute quantification of diGly peptides via AQUA peptides for each site. While this is theoretically possible, it will require extensive optimisation and is, therefore, beyond the scope of the present study. There is also concern that mutation of a Ub acceptor Lys can cause alternative ubiquitylation of a Lys nearby [3, 4]. Thus, we believe that an in-depth analysis of the epistatic relationship between the four ubiquitylated Lys would be best addressed in a separate study.

Supplementary Fig. 4d

Supplementary Fig. 5d

4. The results with reconstituted K219R mutant in *mikl*^{-/-} MDFs vs MDFs from KI K219R/K219R mice show that the reconstitution restores half of the necroptotic response compared to WT but that KI does not restore any necroptosis. Could the authors comment on these discrepancies and is it simply due to different levels in the two different cell line pairs?

We think that this is due to the different expression levels of MLKL in the two systems. The KI scenario (MDFs from KI K219R/K219R mice) enables us to express mutant MLKL from its endogenous locus, achieving physiological levels. Hence, this phenotype will reflect the real effect of the introduced mutation.

In the MDF-reconstitution system, MLKL is inducibly expressed from a transgene. While we have made every attempt to achieve near physiological levels, the achieved expression level of MLKL from the transgene is only an approximation to its normal expression. We have now included this clarification in the revised manuscript (page 11, lines 241-245)

Reviewer #3 (Remarks to the Author):

Nature Communications manuscript NCOMMS-20-32300

“Ubiquitylation of MLKL at lysine 219 positively regulates necroptosis-induced tissue injury and pathogen clearance”, by Ramos Garcia and colleagues.

In this manuscript, the authors report ubiquitylation of MLKL during necroptosis induction by TSZ and TRAILSZ, but not during apoptosis in response to TS. They position the ubiquitylation event downstream of RIPK1/3, subsequent to the phosphorylation of MLKL on S345 by RIPK3, but prior to the plasma membrane localization of MLKL. Using linkage specific ubiquitin pulldowns, they identified K63-linkages as the major constituent of the ubiquitin chains conjugated to MLKL in response to TSZ. Through MS analysis, they then identified 4 lysine residues of MLKL being ubiquitylated following TSZ, and reconstitution of *Mkl1*^{-/-} cells with K>R mutants revealed a major role of K219 for MLKL cytotoxicity under this challenge. To evaluate the physiological relevance of these findings, the authors generated transgenic *Mkl1*K219R mice by CRISPR/Cas9, and report viability of the mice. However, they show that cells derived from these animals are protected from the necroptotic trigger TSZ, and that the mutation alters MLKL oligomerization at the plasma membrane. Finally, and in accordance with their findings, they show that *Mkl1*K219R/K219R MDFs succumb less to MCMV M45mutRHIM infection, and that *Mkl1*K219R/K219R mice are protected from necroptosis-induced injury caused by subcutaneous injection of IAP antagonist + caspase inhibitor.

Comments:

This is a clear and very well written manuscript presenting novel and high-quality results on the importance of K219 ubiquitylation for MLKL cytotoxicity. The conclusions drawn by the authors are supported by solid data. The study therefore provides important new insights on the regulation of TNF-induced necroptosis, and most probably on MLKL-induced necroptosis in general. I however still have some concerns that I believe should be addressed by the authors.

We thank the reviewer for the positive feedback and the suggestions to improve this manuscript.

1) The authors should show the effect of the K219R mutation on the K63-ubiquitylation of MLKL in response to TSZ. Indeed, Figures 1-3 are dedicated to revealing ubiquitylation of MLKL during necroptosis, but there is no data showing that ubiquitylation is reduced in the K219R mutant.

We have followed the reviewer's suggestion and have included our data on the ubiquitylation pattern of *Mkl1*^{K219R}. *Mkl1*^{K219R} showed a reduction of the Ub smearing pattern in cells from knock-in animals (**Supplementary Fig. 4d**). Thus, mutation of K219 reduces K63-linked MLKL ubiquitylation.

Supplementary Fig.4d

2) The authors should not only discuss but also experimentally evaluate the importance of MLKL ubiquitylation on K230 in human cells. This is required to demonstrate the conserved aspect of this regulatory mechanism.

This issue was also brought up by Reviewer 2.

We have expanded our manuscript by highlighting that i) K230 of human MLKL is also ubiquitylated (K230 of human MLKL corresponds to K219 of mouse MLKL), and ii) that mutation of K230 renders human MLKL less active. The functional consequence of hMLKL^{K230M} was previously addressed by Petrie et al [2], demonstrating that hMLKL^{K230M} is significantly less potent in inducing cell death in HT-29 cells.

These data are consistent with the notion that human MLKL is ubiquitylated during necroptosis and that K230 contributes to the cytotoxic potential of human MLKL. However, due to the structural differences between mouse and human MLKL, caution should be exercised when extrapolating to the human setting. We have discussed this point in the Discussion (page 16, lines 369-377)

3) A smear of S345A MLKL is still pulled down by TUBE (ub) upon TSZ stimulation (Fig. 2d). This seems to be contradictory to the statement of the authors that phosphorylation is a prerequisite for ubiquitylation. The authors should perform a DUB treatment to exclude the ubiquitylation nature of the smear.

We have followed the reviewer's suggestion, and have conducted a DUB treatment assay of the experiment shown in Fig. 2d. This demonstrated that MLKL^{S345A} was still ubiquitylated (see faint upper bands (**Supplementary Fig.2d**)). While MLKL^{S345A} was still ubiquitylated, the ubiquitylation pattern of MLKL^{S345A} was significantly different to the one of MLKL^{WT}, where two prominent lower molecular weight bands are the main forms of Ub-MLKL. Consequently, we have rephrased our conclusion.

We now state: "The appearance of the prominent, lower molecular weight ubiquitylation smearing pattern of MLKL is dependent on its phosphorylation at S345, while the faint, higher molecular weight ubiquitylation smearing pattern occurs irrespective of MLKL phosphorylation at S345".

We speculate that MLKL phosphorylation at S345 triggers a conformational change of MLKL that exposes certain Lys residues (such as K219). This leads to appearance of two prominent bands, immediately above unmodified MLKL. Abrogation of MLKL phosphorylation prevents this MLKL modification because these two upper bands completely disappear. However, certain Ub modifications, those that migrate at higher molecular weight, can still occur in the phospho-mutant setting.

Supplementary Fig.2d

Also, how does the authors explain the increased binding of non-modified S345A MLKL to the TUBE upon TSZ in Fig. 2d (not observed in RIPK3i conditions – Fig. 2b-c)? Would it mean that non-ubiquitylated MLKL is bound to other ubiquitylated proteins in response to TSZ? This would give weight to my concerns regarding the PLA results (see below).

This is a general phenomenon that is frequently observed when purifying ubiquitylated proteins, not just for MLKL, but also for other proteins [5]. Either there are residual interactions to the matrix, or non-modified proteins dimerize with ubiquitylated forms.

Despite this, our data clearly demonstrate that the smearing pattern of purified MLKL is indeed due to the conjugation of Ub. This is evident as DUB digestion collapses the smear to a single band that corresponds to unmodified MLKL with its expected molecular weight (**Supplementary Fig. 1d**)

We share the concern regarding the PLA. Since PLA relies on proximity, we cannot rule out the possibility that the obtained signal comes from Ub that is merely proximal but not attached to MLKL (see our response below).

Supplementary Fig.1d

4) It would be of interest to know the identity of the E3 ligase conjugating MLKL with ubiquitin chains. While performing an unbiased screen to identified this E3 is certainly out of the scope of this study, the authors could still test whether Peli1 ubiquitylate MLKL. To my knowledge, Peli1 is indeed the only pro-necroptosis E3 ligase identified so far (PMID:29078411).

We have followed the reviewer's suggestion and tested the role of Peli-1 in regulating MLKL and necroptosis. We found that Peli-1 plays no role in MLKL ubiquitylation or MLKL-mediated necroptosis in MDFs. According, CRISPR-mediated deletion of Peli-1 neither changed MLKL ubiquitylation nor MLKL-mediated necroptosis (**Supplementary Fig. 8 a-c**).

Supplementary Fig.8 a-c

5) It is somehow disappointing that the only *in vivo* data evaluating the importance of K219 ubiquitylation of MLKL are obtained in an artificial and rather irrelevant model (subcutaneous injection of AE). It would have been much more interesting the cross the mice with a genetic model of disease driven by necroptosis. Since the authors already used MCMV M45mutRHIM *in vitro*, they could at least additionally perform *in vivo* infection experiments to further demonstrate the *in vivo* importance of K219 ubiquitylation of MLKL.

Indeed, it would have been interesting to cross these mice to genetic models of necroptosis. Unfortunately, our animal facility is operating at reduced levels during the current COVID-19 pandemic, and we are prevented from conducting long-term experiments due to staff shortage. We already had to terminate several strains due to the restrictions imposed by COVID-19. This prevents us from conducting the suggested genetic experiments.

Also, we currently do not hold the required license to perform *in vivo* MCMV experiments. The time for license approval would be in excess of 6 months. Combined with pilot experiments and troubleshooting, this would easily amount to 8-10 months for the completion of the MCMV study. All *in vitro* viral work was conducted by Jason Upton. Unfortunately, Jason does not hold an animal license.

We would like to re-iterate that the skin injury model faithfully recapitulates skin-specific genetic deletion of cIAP1/cIAP2 [6]. We also would like to highlight that the skin phenotype is entirely due to necroptosis. This is evident as ASTX660/emricasan (AE)-induced skin injury is completely blocked by pharmacological inhibition of RIPK1 (**Supplementary Fig.7 a-i**) and genetic ablation of *Mik1* (**New Figure 7 a-i**).

To strengthen our *in vivo* data, we have increased the cohort size (**New Figure 7 a-i**), and conducted additional experiment with a pharmacological inhibitor of RIPK1 (**Supplementary Fig.7 a-i**). These data demonstrate that AE triggers RIPK1-driven and MLKL-dependent necroptosis.

Lastly, to exclude any basal alterations in the immune landscape of *Mik1^{K219R}* animals, we quantified immune cells of the spleens of *WT*, *Mik1^{KO}*, and *Mik1^{K219R}* animals. As shown in **Supplementary Fig. 5b**, the immune landscape of these animals is normal.

New Figure 7 a-i

Supplementary Fig.7 a-i

Supplementary Fig.5b

Additional comments

6) The authors should provide an explanation for the presence (and sometimes absence) of non-ubiquitylated MLKL in the different ubiquitin pulldowns. Also, the authors write that MLKL is massively ubiquitylated, but this statement may need to be tuned down as it rather looks like only a minor fraction of MLKL is ubiquitylated (for example Fig. 1C).

This issue was already raised above (see point 3)

The fact that non-ubiquitylated forms are also co-purified is a general phenomenon that is frequently observed when purifying ubiquitylated proteins, not just for MLKL, but also for other proteins [5]. Either there are residual interactions to the matrix, or non-modified proteins dimerize with ubiquitylated forms.

We have now rephrased the sentence regarding MLKL ubiquitylation.

7) As MLKL is most probably in complex with other ubiquitylated proteins (such as RIPK1 or RIPK3), the PLA results appear poorly informative. Indeed, they could just reflect non-ubiquitylated MLKL in complex with other ubiquitylated proteins. This possibility should at least be mentioned in the discussion.

We have now modified the discussion section to include this possibility regarding the PLA results (page 15, lines 375-378). We have discussed this concern also in our response to point 3.

8) Fig. 2D, the arrow for P-MLKL does not seem to be at the correct place.

This has now been corrected.

9) How do the authors explain the signal corresponding to non-modified MLKL in Fig. 3C? Non-specific binding only revealed under TSZ?

Under steady-state conditions MLKL is complexed with HSP90, preventing it from adopting a binding competent form. Once active, MLKL undergoes a conformational change, which might make MLKL more prone to artefactual interactions, including interactions with GST.

10) Fig. 7B-C, the staining is very light and the histology not very apparent.

This has now been corrected.

11) Fig. 6D the authors should show MLKL levels by SDS-PAGE in the lysate of each fraction.

This is now included.

12) Line 101-110: The authors claim that ubiquitylation would precede oligomerization. However, the results are only correlative. In order to demonstrate this point, the authors should evaluate the ubiquitylation status of an MLKL mutant that cannot oligomerize.

This has now been addressed. With oligomerisation we were referring to higher order polymers. To test whether higher order oligomerisation is required for ubiquitylation, we have made use of a widely characterized MLKL mutant that is unable to form higher order polymers at the plasma membrane due

to an N-terminal FLAG-tag [7]. As now shown in **Supplementary Fig. 2f**, FLAG-MLKL is phosphorylated and ubiquitylated, suggesting that both posttranslational modifications occur prior to higher order oligomer assembly. Note, the FLAG-MLKL also carries a biotin ligase (AP2) at the C-terminal end of MLKL.

Supplementary Fig. 2f

References

1. Akimov, V., et al., *UbiSite approach for comprehensive mapping of lysine and N-terminal ubiquitination sites*. Nat Struct Mol Biol, 2018. **25**(7): p. 631-640.
2. Petrie, E.J., et al., *Conformational switching of the pseudokinase domain promotes human MLKL tetramerization and cell death by necroptosis*. Nat Commun, 2018. **9**(1): p. 2422.
3. Feltham, R., et al., *Mind Bomb Regulates Cell Death during TNF Signaling by Suppressing RIPK1's Cytotoxic Potential*. Cell Rep, 2018. **23**(2): p. 470-484.
4. Tang, Y., et al., *K63-linked ubiquitination regulates RIPK1 kinase activity to prevent cell death during embryogenesis and inflammation*. Nat Commun, 2019. **10**(1): p. 4157.
5. Emmerich, C.H. and P. Cohen, *Optimising methods for the preservation, capture and identification of ubiquitin chains and ubiquitylated proteins by immunoblotting*. Biochem Biophys Res Commun, 2015. **466**(1): p. 1-14.
6. Anderton, H., et al., *Inhibitor of Apoptosis Proteins (IAPs) Limit RIPK1-Mediated Skin Inflammation*. J Invest Dermatol, 2017. **137**(11): p. 2371-2379.
7. Hildebrand, J.M., et al., *Activation of the pseudokinase MLKL unleashes the four-helix bundle domain to induce membrane localization and necroptotic cell death*. Proc Natl Acad Sci U S A, 2014. **111**(42): p. 15072-7.

REVIEWERS' COMMENTS

Reviewer #2 (Remarks to the Author):

The authors addressed my comments in full. Thank you for the effort. I recommend publication of the revised manuscript in Nature Communication.

Tudor Moldoveanu

Reviewer #3 (Remarks to the Author):

The authors responded to my comments, and I am satisfied by the way they addressed them.

I am just not entirely convinced by the authors' interpretation of the results of SFig.2d, which have been obtained to address my concern about the original statement that phosphorylation is a prerequisite for ubiquitylation. The new set of data show that S345A MLKL is indeed still ubiquitylated, and the authors interpret the difference in the smearing profile with WT MLKL as a reduction in ubiquitylation (disappearance of the low MW forms). However, the disappearance of the low MW forms is in profit of higher MW ubiquitylated forms of MLKL, which may instead suggest increased ubiquitylation. In line with this alternative interpretation, the proportion of non-modified MLKL in response to TSZ is also reduced in the S345A mutant, but restored to WT levels upon USP21 treatment. The authors may decide to discuss this alternative interpretation of the results.

I congratulate the authors for the nice study.

Reviewer #4 (Remarks to the Author):

The manuscript entitled 'Ubiquitylation of MLKL at lysine 219 positively regulates necroptosis-induced tissue injury and pathogen clearance' by Ramos Garcia and co-authors describes an effort to describe the role of Lys 219-ubiquitination of MLKL protein function in necroptosis.

I have carefully read authors' response to previous comments, and I'm glad to see the changes and improvements the authors have done for this matter. The manuscript is very well written in general, with quality results nicely presented. I am particularly satisfied with how mass spectrometry experiments were conducted and the results that came out of it. I am particularly puzzled with how the TMT experiment worked well, with using only 20 mg of proteins per sample, following all the additional steps (protein precipitation, digestion, IP and sample cleaning) where the loss of proteins and peptides can be significant. Did authors measure the TMT-labeled peptide concentration from each sample before combining them into a final sample? This is very important, as the samples went through a lot of steps before being ready for MS analysis. What was the amount of peptides injected for LC-MS/MS? Otherwise, excellent work.

Response to the reviewers: manuscript NCOMMS-20-32300**REVIEWERS' COMMENTS****Reviewer #2 (Remarks to the Author):**

The authors addressed my comments in full. Thank you for the effort. I recommend publication of the revised manuscript in Nature Communication.

Tudor Moldoveanu

We would like to thank the Reviewer 2 for the time and effort dedicated to the revision of this manuscript.

Reviewer #3 (Remarks to the Author):

The authors responded to my comments, and I am satisfied by the way they addressed them.

I am just not entirely convinced by the authors' interpretation of the results of SFig.2d, which have been obtained to address my concern about the original statement that phosphorylation is a prerequisite for ubiquitylation. The new set of data show that S345A MLKL is indeed still ubiquitylated, and the authors interpret the difference in the smearing profile with WT MLKL as a reduction in ubiquitylation (disappearance of the low MW forms). However, the disappearance of the low MW forms is in profit of higher MW ubiquitylated forms of MLKL, which may instead suggest increased ubiquitylation. In line with this alternative interpretation, the proportion of non-modified MLKL in response to TSZ is also reduced in the S345A mutant, but restored to WT levels upon USP21 treatment. The authors may decide to discuss this alternative interpretation of the results.

I congratulate the authors for the nice study.

We thank Reviewer 3 for his/her time in reviewing this manuscript. We have expanded our discussion as suggested:

'The prominent low molecular weight ubiquitylation events on MLKL were dependent on RIPK3-mediated phosphorylation of MLKL at S345. Interfering with MLKL phosphorylation, either via pharmacological inhibition of RIPK1 and RIPK3, or expression of a phospho-mutant form of MLKL (MLKL^{S345A}), abrogated MLKL phosphorylation and Ub modifications of low molecular weight. Intriguingly, while the low molecular weight ubiquitylation events were phospho-dependent, higher molecular weight modifications appeared to be slightly elevated upon inhibition of S345 phosphorylation.'

Reviewer #4 (Remarks to the Author):

The manuscript entitled 'Ubiquitylation of MLKL at lysine 219 positively regulates necroptosis-induced tissue injury and pathogen clearance' by Ramos Garcia and co-authors describes an effort to describe the role of Lys 219-ubiquitination of MLKL protein function in necroptosis. I have carefully read authors' response to previous comments, and I'm glad to see the changes and improvements the authors have done for this matter. The manuscript is very well written in general, with quality results nicely presented. I am particularly satisfied with how mass spectrometry experiments were conducted and the results that came out of it. I am particularly puzzled with how the TMT experiment worked well, with using only 20 mg of proteins per

sample, following all the additional steps (protein precipitation, digestion, IP and sample cleaning) where the loss of proteins and peptides can be significant. Did authors measure the TMT-labeled peptide concentration from each sample before combining them into a final sample? This is very important, as the samples went through a lot of steps before being ready for MS analysis. What was the amount of peptides injected for LC-MS/MS? Otherwise, excellent work.

We thank the reviewer for the positive comments with regards to the manuscript, and the mass spectrometry experiment in particular.

First, we took 10 % of the samples for label-free analysis to test each sample. The respective LC-MS/MS traces looked very similar, being in the range in which the TMT normalisation can be adjusted. This is because the sample amount was low, and we couldn't afford to perform a peptide assay. Then we used the remaining 90 % for TMT labelling. We used plenty of TMT reagent, so the labelling efficiency was satisfactory. For example, for Fraction 4 (P04) (by setting TMT6plex on peptide N-term as variable instead of fixed) the labelling efficiency was 98.6%.

We believe that our label-free test was a careful step to check the samples before the TMT labelling. We are not aware that a peptide assay should be performed prior to TMT labelling, but we thank the reviewer for this suggestion. To our knowledge, Steven Gygi's lab is the only lab that performs this in the suggested manner. We will take this into consideration for our future studies.